

# Megadunes in Antarctica: migration and evolution from remote and in situ observations

Giacomo Traversa[1,2], Davide Fugazza[2] and Massimo Frezzotti[3]

[1] Department of Physical Sciences, Earth and Environment (DSFTA), Università degli Studi di Siena, 53100 Siena, Italy
[2] Department of Environmental Science and Policy (ESP), Università degli Studi di Milano, 20133 Milan, Italy
[3] Department of Science, Università degli Studi Roma Tre, 00146 Rome, Italy

*Correspondence to*: Giacomo Traversa (giacomo.traversa@student.unisi.it) and Massimo Frezzotti (massimo.frezzotti@uniroma3.it)

**Abstract.** Megadunes are peculiar snow dune fields known to be present only on the East Antarctic plateau and other planets (Mars and Pluto). Antarctic megadunes are climatically important because their leeward flanks are characterized by glazed surfaces, a particular morphogenetic state of snow which makes these zones ablation areas, as their surface mass balance is near-zero or negative, on a continental ice sheet where surface mass balance is on average positive. This work builds on previous efforts in this field and by taking advantage from the most recent remote-sensing products and techniques coupled with field data, aims to provide new information and confirm previous hypotheses about megadunes. Focusing on two sample areas of the East Antarctic plateau where in the past international field activities were carried out (EAIIST and It-ITASE), we analysed the dynamic parameters of megadunes, their albedo and morphology. For the first time we provide a detailed analysis of their upwind migration, in all its components (absolute, sedimentological and ice flow) from remote and field observations, finding absolute values of approximately 10 m a$^{-1}$ and demonstrating the upwind migration of dune windward flanks, with a relative stability of the leeward faces. Using remote sensing, we analysed their optical characteristics, i.e., albedo (broadband and NIR), brightness temperature and topographic parameters, including slope, aspect and slope along the prevailing wind direction (SPWD). First numerical results about glazed-surface albedo are thus provided, which is found to be lower than the surrounding snow, especially in NIR wavelengths. This detailed information allowed us to perform a precise mapping of glazed surfaces and their evolution and trends over time, demonstrating a general overall intra-annual areal decrease in summer (-16%) and an inter-annual increase over recent years (at maximum almost +0.2% per year in January).

## 1 Introduction

Antarctic climate and mass balance have been highlighted by the Special Report on the Ocean and Cryosphere in a Changing Climate (Meredith et al., 2019) by the Intergovernmental Panel on Climate Change (IPCC) among the main uncertainties for the climate system and sea level projections. Surface mass balance (SMB) is the net balance between the processes of snow accumulation and loss on a glacier surface and provides mass input to the surface of the Antarctic Ice Sheet. Therefore, it represents an important control on ice sheet mass balance and resulting contribution to global sea level change. Ice sheet SMB



varies greatly across multiple scales of time (hourly to decadal) and space (meters to hundreds of kilometres), and it is
notoriously challenging to observe and represent in atmospheric models. Moreover, given the difficulties in accessing the
interior of the ice sheet, only limited field observation on past and current conditions exists. The part of the East Antarctic Ice
Divide closest to the South Pole is the coldest and driest area on Earth and presents several unique features (e.g. megadunes,
glazed surfaces etc.) and processes (e.g. extensive post-depositional effects in the snow as well as interactions between the
snow surface and the overlying atmosphere) that remain relatively unexplained (Fahnestock et al., 2000; Frezzotti et al., 2002a,
b; Courville et al., 2007; Scambos et al., 2012). The glazed surface or wind crust areas show a local net SMB close to zero, in
contrast with the rest of the continent where SMB is mainly positive (Agosta et al., 2019), which leads to characteristics of the
surface that can be observed from the satellites (Fahnestock et al., 2000; Scambos et al., 2012). Megadune morphologies form
through the interaction of peculiar conditions of snow accumulation and metamorphism, wind, and topography. Previous works
have already studied megadune fields in Antarctica using ground penetration radar (GPR; Frezzotti et al., 2002a, b; Arcone et
al., 2012a, b; Das et al., 2013; Ekaykin et al., 2015), atmospheric models (Dadic et al., 2013; Palm et al., 2011, 2017) and
remote sensing, both at high resolution (Frezzotti et al., 2002a, b; Traversa et al., 2021a), based on previous satellites of the
Landsat family, and lower spatial resolution by using other images (Das et al., 2013), such as Radarsat/MODIS (Fahnestock et
al., 2000; Arcone et al., 2012a; Scambos et al., 2012) and AVHRR (Fahnestock et al., 2000). However, a detailed analysis
of megadune physical parameters and their migration over time is still lacking.
This study focuses mainly on satellite image analysis of two megadune areas of the East Antarctic Ice Sheet (EAIS) situated
respectively 300 km East of Vostok Station and 150 km East of Concordia Station (both areas are located in the same huge
megadune field, approximately 450,000 km$^2$, Fahnestock et al., 2000, Fig.1), characterized by the presence of megadune
formations. Both areas were crossed and surveyed by two snow traverses respectively in the 2018-19 EAIIST expedition (East
Antarctic International Ice Sheet Traverse) and 1998-1999 It-ITASE (Italian-International TransAntarctic Scientific
Expedition) expedition. The analysis of survey data of the first area (EAIIST, 300 km East of Vostok Station,
https://www.eaiist.com/en/) are under process whereas the in situ observations of the second traverse It-ITASE are available
(Frezzotti et al., 2002a, b, 2004, 2005; Proposito et al., 2002; Vittuari et al., 2004).





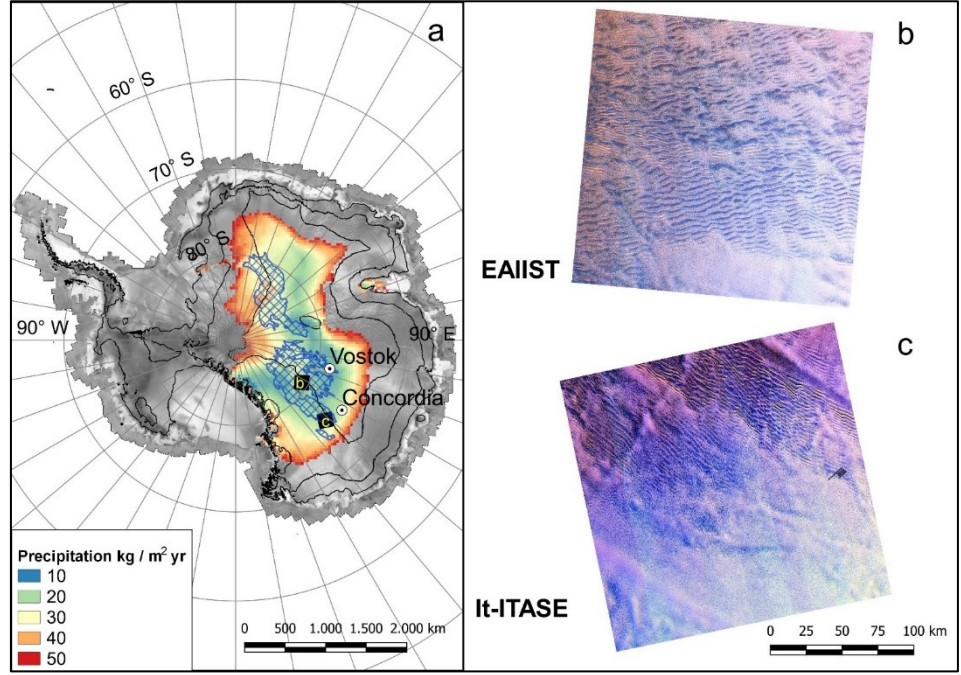

**Figure 1: Antarctic continent with the zoom (see b and c in panel a) of the study areas, i.e., EAIIST (b) and It-ITASE (c, with the**
**traverse transects in black) tile of Landsat (false colour composites). In the background (a): surface elevation contour lines every**
**1000 m, RAMP Radarsat Mosaic (Jezek, 1999), RACMO total precipitation rate (Van Wessem et al., 2014) and megadune fields as**
**blue reticulate polygons (Fahnestock et al., 2000). Polar Stereographic projection.**

The aim of the study is to provide a detailed survey on megadunes of the Antarctic continent by remote sensing, considering
different physical parameters captured through Landsat 8 OLI satellite images at 15/30 m spatial resolution. Our study focuses
on two Landsat tiles, i.e. 069119 (EAIIST site, Lat 80°47' S Long 122°19' E) and 081114 (It-ITASE site, Lat 75°54' S Long
131°36' E), in order to provide new information about different parameters of megadune fields: dune migration, albedo
(spectral and broadband), brightness temperature and morphologies. A mapping of glazed surfaces associated to these
megadune fields is provided, based on the above-mentioned parameters.

## 1.1 Megadune fields

Megadunes were first described by Swithinbank (1988), who termed these features based on their likeness to transverse sand
megadunes on satellite images. They rise a few meters above the general surface level and are imperceptible to the surface
traveller. When viewed from an airplane or satellite, these features appear as long waves of alternating blueish (glazed surface)
and white bands (sastrugi). The surface waveforms with regular bands of sastrugi and glazed surfaces allow surveying the
megadunes by satellite observations, because of differences in albedo (Frezzotti et al., 2002a) and microwave backscatter
(Fahnestock et al., 2000). These features are widespread in the remotest areas of the southern part of the East Antarctic Ice
Divide and extend for more than 500,000 km² (Fahnestock et al., 2000; Frezzotti et al., 2002a), as a result of uncommon snow



accumulation and redistribution processes; thus, they have a crucial role in SMB and ice core interpretation. In fact, in such inland areas, the stability of climatic conditions could play a key role in megadune formation, since accumulation is very low while katabatic wind intensity and direction are stable; these conditions could affect snow sintering and a high grade of snow
metamorphism  (Albert et al., 2004; Courville et al., 2007; Scambos et al., 2012; Dadic et al., 2013). The climatic conditions of megadune fields are characterized by extremely low temperatures (mean annual temperatures from –45° to –60°C), extremely low snow accumulation (<50 mm water equivalent per year, w.e.a$^{-1}$). Megadunes are oriented perpendicular to the slope along the prevailing wind direction (SPWD), wave amplitudes are small (about 1-4 m), wavelengths range from 2 to over 5 km with a 3 km average (amplitude and wavelength statistics are here calculated using almost fifty megadunes sampled
in the transects at the EAIIST site, Fig. 2), and megadune crests are nearly parallel, extending  from tens to hundreds of kilometres (Swithinbank, 1988; Fahnestock et al., 2000; Frezzotti et al., 2002a, b; Arcone et al., 2012a, b). The angle between the wind direction (50° – 60°) and the direction of general surface slope (95° - 100°) at a regional scale is about 40° - 50° at It-ITASE (Frezzotti et al., 2002b) and about 30° - 50° at EAIIST with 40° - 50° of wind direction and 80° - 85° of aspect. This clearly shows the katabatic flow draining from the high plateau and turning to the left under the action of the Coriolis force.
For these reasons, the directions of megadunes and sastrugi are fundamental in order to evaluate the local orientation of prevalent wind direction (Mather, 1962; Parish and Bromwich, 1991; Fahnestock et al., 2000). At kilometre scale, regional SPWD in the megadune area was calculated by Frezzotti et al. (2002b) as 0.10 – 0.15% (1.0 – 1.5 m km$^{-1}$), less than half of the megadune leeward-face slope of 0.4–0.5% (4–5 m km$^{-1}$).

Based on previous studies, the SMB range is 25% (leeward faces, characterized by the presence of wind glazed surfaces, as
described later) and 120% (windward faces, covered by huge sastrugi up to 1.5 m height) of accumulation in adjacent non-megadune areas (Frezzotti et al., 2002b). In addition, the sedimentary structure of buried megadunes examined via ground penetrating radar (GPR) suggests that the sedimentary morphology of the windward face (sastrugi) migrates upwind with time, burying the glazed surface of the leeward face (Frezzotti et al., 2002b; Ekaykin et al., 2015), in contrast with sand dunes (which show a downward migration) and typical of "antidune" formations of fluvial and ocean bedforms (Prothero and Schwab, 2004).
This uphill migration is caused by the difference in  accumulation between windward (high accumulation) and leeward (near-zero accumulation) sides, also leading  to differences in snow features and surface roughness (Fahnestock et al., 2000; Albert et al., 2004; Courville et al., 2007). According to Dadic et al. (2013), who based his analysis on superficial-flow theory for sediments in water (Núñez-González and Martín-Vide, 2011) and atmospheric flow modelling, persistent katabatic winds, strong atmospheric stability and spatial variability in surface roughness are the primary controllers of upwind accumulation
and migration of megadunes, where the latter represents the main factor that influences their velocity. In association with megadunes, wide glazed surfaces are present in the region where the SPWD is higher than 0.25° (0.4% or 4 m km$^{-1}$). Glazed-surface area shows deep cracks that are more evident in the early summer season (Watanabe, 1978; Frezzotti et al., 2002a; Albert et al., 2004; Courville et al., 2007), caused by repeated thermal cycles (thermal expansion and contraction of snow or firn layers), and patterns with polygonal forms, correlated to a long-term hiatus in accumulation of snow (Watanabe, 1978).
Fortunately, the spectral differences between glazed areas and surrounding snow make their identification easier from space,





as the former have a lower albedo (Fujii et al., 1987; Frezzotti et al., 2002a; Scambos et al., 2012). Spectral differences also lead to an effect on temperatures, which is on average higher than on the snow surface (Fujii et al., 1987). Megadunes appear to be formed by an oscillation in the katabatic air flow, leading to a wave-like variation in net accumulation. The wind-waves are formed at the change of SPWD, in response to the buoyancy force, favouring the standing-wave mechanism (Fahnestock et al., 2000; Frezzotti et al., 2002b).

## 2 Materials and Method

### 2.1 Materials

In order to study the megadunes and glazed areas, we used Landsat 8 OLI satellite images from 2013 to 2021 (https://earthexplorer.usgs.gov/). Landsat 8 provides data in several spectral bands, including visible (0.45-0.67μm, bands 2-3-4) near infrared (NIR, 0.85-0.88 μm band 5) and short-wave infrared (1.57-2.29 μm bands 6-7), at 30 m spatial resolution, with three more bands at different spatial resolution: a panchromatic band (0.503-0.676 μm, 15 m, band 8) and two thermal infrared bands (10.6-12.51 μm, 100 m, bands 10-11). In this study, we used all the VIS and IR bands (from 2 to 7) to calculate albedo and map megadunes, thermal bands to calculate brightness temperature and the panchromatic band to perform the megadune migration analysis. Excluding from our dataset all images with cloud cover > 10% of land surface and visible blowing snow events, we obtained 21 images from Landsat 8 during the period of satellite observation (Table A1), 14 for the EAIIST site and 7 for It-ITASE. Cloud cover was detected using the value reported in the Landsat Metadata, which is determined by the CFMask (C code of the Function of Mask) algorithm (Foga et al., 2017) and, since it may be inaccurate over bright targets as snow, an additional visual check for each analysed image was performed. For the calculation of megadune migration (Sect. 2.2.4), also Sentinel-2 optical imagery (Band 8 NIR) was used. Sentinel-2 images were acquired using the same criteria as for Landsat, with the advantage of having a higher spatial resolution of 10 m.

In addition, we extracted wind direction from ERA5 (Hersbach et al., 2020) and by identification of sastrugi based on Landsat (Sect. 2.2.3). ERA5 is distributed by the European Centre for Medium-Range Weather Forecasts (ECMWF) and is an atmospheric reanalysis global climate dataset. In particular, we used *hourly data* (DOI: 10.24381/cds.adbb2d47). The wind speed and direction were obtained by combining 10 m $u$ and $v$ wind components, and averaged on a 20-year temporal period, from 2000 to 2019. In ERA5 $u$ and $v$ are calculated as:

$$u = -\left|\vec{V}\right| \sin \phi$$
$$v = -\left|\vec{V}\right| \cos \phi \qquad (1)$$
$$\vec{V} = \sqrt{u^2 + v^2}$$

and $\phi$ is the meteorological wind direction (direction from where the wind is blowing, with 0° = N and increasing clockwise). Beside using all wind speed observations, we further divided wind speed in 5 classes, only considering wind speed values above specific thresholds: >3 m s$^{-1}$, >5 m s$^{-1}$, >7 m s$^{-1}$ and >11 m s$^{-1}$. These thresholds were chosen based on the interactions



between wind and snow: snow transportation by saltation (within 0.3 m in elevation) starts at wind speeds between 2 and 5 m $s^{-1}$, transportation by suspension (drift snow) starts at velocities > 5 m $s^{-1}$ (within 2 m) and blowing snow (snow transportation higher than 2 m) starts at velocities between 7 and 11 m $s^{-1}$ (see Frezzotti et al., 2004 and references therein). The threshold

wind speed at which the sublimation of blowing snow starts to contribute substantially to katabatic flows in a feedback mechanism appears to be 11 m $s^{-1}$ (Kodama et al., 1985; Wendler et al., 1993).

Finally, in order to obtain aspect and slope of the surface for the SPWD calculation and perform topographic correction for the calculation of albedo, we used the Reference Elevation Model of Antarctica (REMA, Howat et al., 2019) digital elevation model (DEM; www.pgc.umn.edu/data/rema/). This DEM provides the first high spatial resolution (8 m) terrain map using

hundreds of thousands of individual stereoscopic pairs of submeter (0.32 and 0.5 m) DigitalGlobe satellite imagery. Each individual DEM was vertically registered to satellite altimetry measurements from Cryosat-2 and ICESat, resulting in absolute uncertainties of less than 1 m. It is based mainly on 2015-2016 imagery acquired during the austral summer period (December-March) (Howat et al., 2019). At the EAIIST site, the temporal period is slightly wider (from 2008 to 2017), although 86% of stripes were acquired in 2013-2017 (Table A2).

**2.2 Methods**

Considering our aims, the study includes four main processing steps: Landsat 8 OLI image processing for the calculation of spectral and broadband albedo; extraction of brightness temperatures from Landsat thermal bands; SPWD calculation from ERA5 (Hersbach et al., 2020) and sastrugi-based wind direction; estimation of the superficial speed of megadunes and the direction and intensity of their migration over time.

To characterize wind glazed areas and megadunes, we chose dates that corresponded with the first stripe acquisitions of the REMA DEM (2013, Table A2). We further created 7 sample transects (5 in the EAIIST area and 2 at It-ITASE) (Fig.2) and 30 polygons (equally subdivided in the EAIIST and It-ITASE areas) and performed statistical calculations to identify thresholds of albedo, brightness temperature and SPWD to discriminate between glazed surfaces and surrounding snow. The seven transects are in different areas of the megadune field and they show different wind directions, with the aim of

representing the widest possible range of SPWD values. The transect plots in the paper refer to weighted moving averages based on 11 transect pixels. Each plot is associated to a normalized plot that presents also detrended topography to remove the effect of the topographic slope. Finally, we used Pearson's correlation coefficient to determine the strength of the relationship between the above-mentioned parameters.



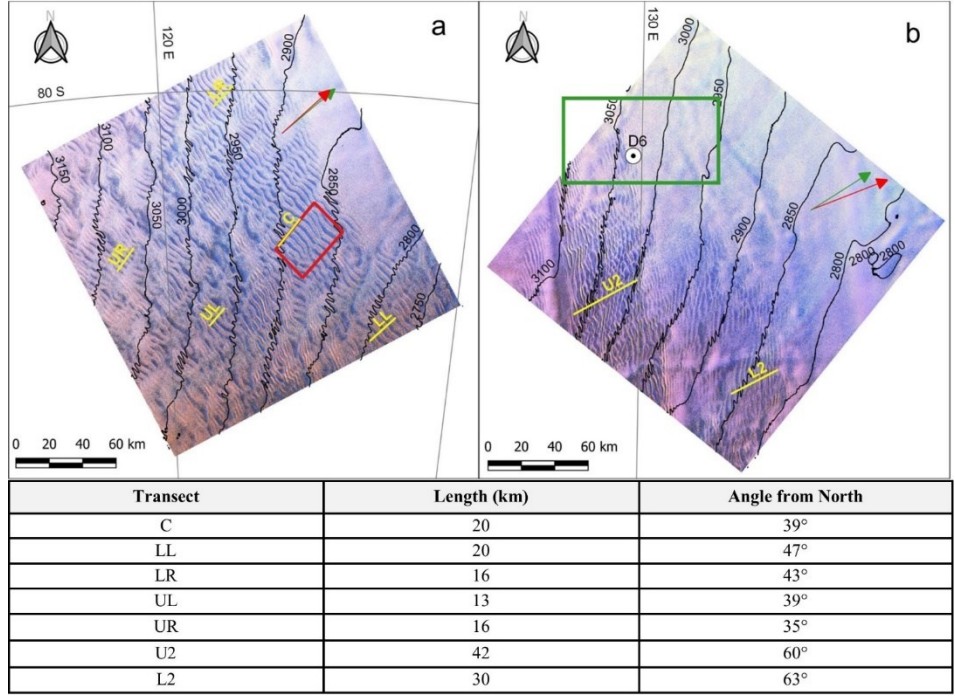

| Transect | Length (km) | Angle from North |
|----------|-------------|------------------|
| C | 20 | 39° |
| LL | 20 | 47° |
| LR | 16 | 43° |
| UL | 13 | 39° |
| UR | 16 | 35° |
| U2 | 42 | 60° |
| L2 | 30 | 63° |

**Figure 2: Locations of transects (yellow) on the false colour images of the EAIIST site (a, Lat 80°47' S Long 122°19' E) and It-ITASE site (b, 75°54' S Long 131°36' E). Red rectangle (a) is the area for the analysis of variations of glazed surfaces (Fig.7) and green rectangle (b) is the zoom in of Fig.3a; green arrows ERA5 wind direction and red arrows sastrugi-based wind direction. Transect length and angle from North (lower panel). Universal Transverse Mercator (UTM) projections.**

### 2.2.1 Spectral and broadband Albedo calculation

The albedo (α), also called bi-hemispherical reflectance, is defined as the ratio of the radiant flux reflected from a unit surface area into the whole hemisphere to the incident radiant flux of hemispherical angular extent (Schaepman-Strub et al., 2006) in the approximate spectral range 350–3000 nm (Bishop et al., 2011). In this study the albedo, both spectral and broadband, was estimated using Landsat 8 OLI imagery, following the method firstly proposed by Klok et al. (2003), then adjusted by Fugazza et al. (2016) and recently tested and validated in Antarctica by Traversa et al. (2021b). In order to obtain the albedo of the sample area, Landsat images need to be processed and corrected. In fact, originally Landsat provides quantized digital numbers (DN, dimensionless) that need to be scaled back to radiance and then converted to Top of Atmosphere (TOA) reflectance. Here, TOA reflectance conversion was performed by using the solar zenith band (obtained from the Angle file of Landsat 8, see Landsat 8 Data User Handbook, Zanter, 2019). This conversion allows applying a more accurate per-pixel correction for the solar zenith angle (SZA), useful in our study considering the average high SZA (>60°), typical of inner Antarctica, and its strong effect on albedo (Pirazzini, 2004; Picard et al., 2016; Traversa et al., 2019). TOA reflectance was then corrected for atmospheric and topographic effects (Klok et al., 2003; Fugazza et al., 2016; Traversa et al., 2021b). For the atmospheric





correction, we used the 6S radiative transfer code (Vermote et al., 1997), using the *i.atcorr* tool of GRASS GIS with an aerosol optical thickness (AOT) at 550 µm of 0.02, based on the values from the AERONET Aerosol Robotic Network

(https://aeronet.gsfc.nasa.gov/cgi-bin/draw_map_display_aod_v3) at Concordia Station (the closest station to our study areas). As regards topographic correction, we used the *i.topo.corr* tool of GRASS GIS with the c-factor method. During these last two steps, the REMA DEM was resampled to 30 m using bilinear interpolation. For a more detailed description of these correction steps, refer back to Traversa et al. (2021b). Finally, narrowband to broadband albedo conversion was carried out by using Liang algorithm (Liang, 2001), applied on bands 2-4-5-6-7 of Landsat 8. The algorithm is:

$$\alpha = 0.356\alpha_2 + 0.130\alpha_4 + 0.373\alpha_5 + 0.085\alpha_6 + 0.072\alpha_7 - 0.0018 \tag{2}$$

where $\alpha_x$ is the spectral albedo of each band and x the band number.

### 2.2.2 Brightness temperature calculation

As regards the retrieval of temperature, we employed the thermal bands of Landsat 8 OLI, i.e. bands 10 and 11. To estimate the TOA brightness temperature received at the satellite, spectral radiance in the thermal bands was converted using the thermal

constants in the Landsat metadata. The TOA brightness temperature is calculated as follows:

$$T = \frac{K_2}{ln\left(\frac{K_1}{L_\lambda}+1\right)} \tag{3}$$

where T is the TOA brightness temperature in Kelvin, $L_\lambda$ the TOA spectral radiance and $K_{1,2}$ the band specific thermal conversion constants from the Landsat Metadata.

### 2.2.3 Calculation of Slope along the prevailing wind direction (SPWD)

The SPWD is one of the most important parameters that characterizes megadune fields (Frezzotti et al., 2002b) and depends on wind direction and surface slope. As regards the first parameter, i.e. wind direction, we extracted it from ERA5. Since ERA5 has a low spatial resolution (30 km), we also calculated sastrugi directions to derive wind orientation, as they are parallel to it (Mather, 1962; Parish and Bromwich, 1991; Fahnestock et al., 2000; Frezzott et al., 2002a and b). Sastrugi-based wind directions were used to validate ERA5 values and were extracted from Landsat 8 OLI at 30 m spatial resolution. In a first step,

we generated False Colour Composites (FCC) of OLI bands 3, 4 and 5 (Green, Red and NIR respectively) which allowed observing differences between snow, glazed surfaces, dunes and sastrugi.

Subsequently, in order to highlight the differences among these morphologies and better identify the sastrugi, we applied a Principal Component Analysis (PCA) on the FCC (*i.pca* module in GRASS GIS, Richards and Richards, 1999). Then, on the first principal component raster, where Band 5 NIR already explained over 90% of the variance on average (solar radiation is

strongly absorbed in this part of the spectrum on glazed surfaces and, in contrast with the visible bands, here feature differences are highlighted, Frezzotti et al., 2002b), we applied an edge detection algorithm (*i.edge* in GRASS GIS, Canny, 1986), with





the aim of excluding other features and highlight sastrugi only. This process was applied on 7 Landsat scenes from the spring, summer and fall months i.e., October, November, December and January of the period 2014-2020). Only small differences were found in these scenes, confirming the stability in direction of sastrugi landforms and thus of katabatic winds. Finally, to
obtain a complete map of wind direction, comparable to the one of ERA5, we applied surface interpolation.

To further calculate the SPWD based on the wind direction from ERA5 and Landsat-derived sastrugi, we implemented a specific algorithm. For each pixel, the SPWD was calculated by taking one of the adjacent pixels to calculate the altitude difference, following the wind direction, following Eq. (4). The algorithm was applied to ERA5 and sastrugi-based wind directions resampled at 120 m spatial resolution.

$$SPWD(i,j) = 1000 \frac{(h(i+k,j+l) - h(i,j))}{d} \tag{4}$$

where $h(i,j)$ is the elevation of a given pixel, and k and l are coefficients $\in \{-1,0,1\}$ used to select the adjacent pixel along the wind direction. k = -1 for NW, N and N, zero for E and W and +1 for SW, S and SE. l = -1 for NW, W and SW, 0 for N and S and +1 for NE, E and SE. $d$ is calculated as the x-y distance between two pixels. The resulting SPWD has units of m km$^{-1}$. The choice of using 2 pixels was taken to avoid an excessive flattening of the SPWD, that would be obtained using a more
conventional slope algorithm (using 9 pixels).

### 2.2.4 Megadune movement

Megadune morphology advances windward whereas the ice flows downhill; the two processes present different directions and modules (Frezzotti et al., 2002a).

By using different images and available data on megadune fields, we are thus able to provide megadune-migration components:
ice-flow (*If*) direction, which is correlated to topographic slope, sedimentological migration (*M_s*), caused by sedimentological process linked to deposition (on the upstream dune flank) and ablation (on the downstream dune flank) of snow, and the result of these processes, the absolute migration (*M_a*):

$$\overrightarrow{M_a} = \overrightarrow{M_s} + \overrightarrow{If} \tag{5}$$

During the 1998-99 It-ITASE traverse, at D6 site (75° 26' 53'' S, 129° 49' 39'' E) an ice velocity of 1.46±0.04 m s$^{-1}$ with an
aspect of 97°(Vittuari et al., 2004) was measured using Global Positioning System (GPS) (It-ITASE site, Fig.3a).



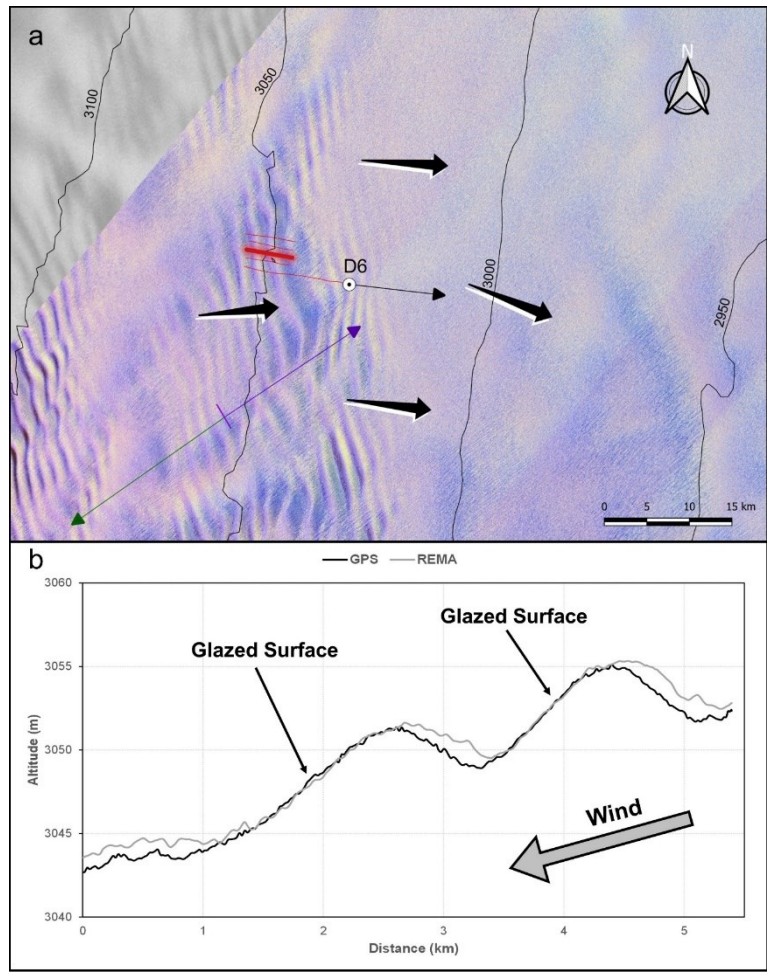

**Figure 3: a) Location of the transects (red) at the It-ITASE site. False colour Landsat image and RAMP Radarsat Mosaic (Jezek, 1999) in background, black arrows as ice-flow (MEaSUREs and D6 site), purple arrow as wind direction from sastrugi and green arrow as dune absolute migration. Universal Transverse Mercator (UTM) projection. b) Topographic section of two megadunes (red highlighted transect in a), with the black line representing elevation from in situ GPS observations (1999) and grey line from REMA DEM (2014).**

Additionally, during this traverse, megadunes were surveyed by means of GPS and Ground Penetrating Radar (GPR) to measure the surface elevation and internal layering of present and buried megadunes; we compared these measurements with the REMA DEM derived by satellite images acquired in 2014 to estimate the sedimentological migration of the megadunes. To estimate the absolute megadune migration, we used Landsat 8 OLI imagery, in particular its panchromatic band (B8), with a spatial resolution of 15 m, and Sentinel-2 imagery (Band 8 NIR) at 10 m spatial resolution. We considered the widest temporal interval between two cloud-free images of Landsat 8 / Sentinel-2, that were in a similar period of the year, in order to avoid





relevant differences in the SZA that could confound the feature tracking algorithm. We identified three pairs of images per area, subdivided in four Landsat pairs and two Sentinel pairs (Table 1).

| Zone | Satellite | $t_0$ | $t_1$ | t span (a) | Mean $M_a$ (m a$^{-1}$) | STD $M_a$ (m a$^{-1}$) | Yearly ERR $M_a$ (m a$^{-1}$) | Features |
|---|---|---|---|---|---|---|---|---|
| It-ITASE | L8 | 02-Dec-2014 | 30-Nov-2019 | 5 | 14.0 | 3.9 | 2.6 | 30073 |
| It-ITASE | L8 | 02-Dec-2014 | 02-Dec-2020 | 6 | 12.8 | 3.4 | 2.5 | 30538 |
| It-ITASE | S2 | 13-Dec-2016 | 27-Dec-2020 | 4 | 11.4 | 3.8 | 1.7 | 537304 |
| EAIIST | L8 | 27-Dec-2013 | 28-Dec-2019 | 6 | 11.9 | 3.6 | 8.9 | 316951 |
| EAIIST | L8 | 17-Dec-2015 | 30-Dec-2020 | 5 | 14.2 | 3.4 | 16.3 | 139622 |
| EAIIST | S2 | 10-Jan-2018 | 02-Jan-2021 | 3 | 10.5 | 4.1 | 11.7 | 1329648 |

**Table 1. Results of the absolute migration of megadunes calculated from IMCORR based on Landsat 8 OLI (L8) and Sentinel-2 (S2) imagery at the It-ITASE and EAIIST sites.**

In addition, we tried performing a long-term analysis, comparing Landsat 7 ETM+ imagery against Landsat 8 OLI (Table A1). The time span is respectively for the two areas as follows: 29-Dec-1999 / 27-Dec-2013 and 29-Dec-1999 / 28-Dec-2019 (EAIIST), and 02-Jan-2000 / 06-Jan-2016 and 02-Jan-2000 / 17-Jan-2020 (It-ITASE). In fact, even if the co-registration error

could be higher for these images as they were acquired from different satellites, the errors should be mitigated by using image pairs acquired from the same path/row in a similar period (i.e. same incidence angle), so that most of the uncertainty arising from both sources cancel out when measuring displacement (Jeong and Howat, 2015).

With the aim of calculating the superficial velocity and direction of megadunes, the feature tracking module *IMCORR* (Fahnestock et al., 1992; Scambos et al., 1992) was run in SAGA GIS . This algorithm performs image correlation based on

two images providing the displacement of each pixel between the second and first image (Jawak et al., 2018). Finally, by dividing the displacement values by the corresponding time period, we obtained the absolute migration of the megadunes in m a$^{-1}$.

For comparison with the previous results, we also employed another method to evaluate the megadune migration. By using Landsat 8 OLI imagery, similarly to what already done for the detection of sastrugi, and applying an edge detection on a FCC

of bands 3, 4 and 5 of Landsat 8, it is possible to identify the edges between leeward and windward zones of megadunes and thus the megadune ridges, since wind glazes and surrounding areas behave differently from the spectral point of view (Frezzotti et al., 2002b). To investigate variations over a period of 20 years, we used Landsat 7 ETM+ images also in this case, creating a FCC with bands 2, 3 and 4. The obtained direction raster was manually cleaned from errors and artefacts (based on size and direction), and then vectorized after a thinning step. Comparing the obtained velocity fields in different years, we could observe





the absolute migration of the dunes. The results from IMCORR and GPS observations were compared with the *MEaSUREs ice-flow velocity* product (Rignot et al., 2017), that provides the highest-resolution (450 m) digital mosaics of ice motion in Antarctica (assembled from multiple satellite interferometric synthetic-aperture radar systems, mostly between 2007-2009 and 2013-2016), showing for each pixel the direction and the velocity of ice flow.

## 3 Results

**3.1 NIR and broadband albedo**

As an initial approach to the spectral analysis of megadune fields, we evaluated its response in the NIR part of the spectrum (OLI Band 5), because it is in such wavelengths that major differences between snow and ice are found (Warren, 1982). Glazed surfaces show an intermediate spectral reflectance between snow and ice (Frezzotti et al., 2002a; Scambos et al., 2012) and thus are easier to detect in the NIR than at visible wavelengths.

We first focused on an intra seasonal analysis of NIR albedo, using 4 scenes between November and January 2013-2014. A general pattern of NIR albedo is detected (see Fig.4a), with higher values in the upwind area and lower ones downwind.

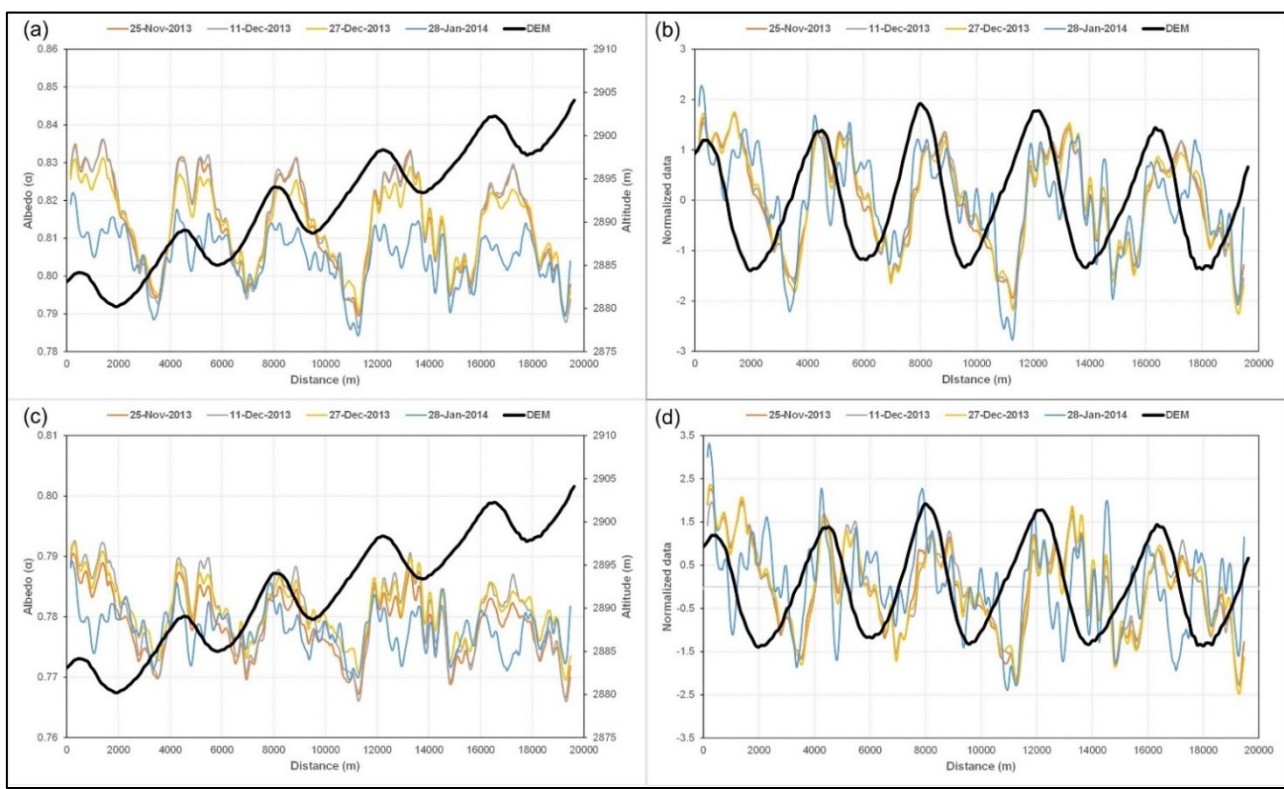

**Figure 4: NIR (a-b) and broadband albedo (c-d) during the austral summer season 2013-2014 for transect C at the EAIIST site (see Fig.2 for location) and elevation from REMA DEM. a and c represent the moving average of 11 pixels and b-d the normalized**
**moving average (with detrended topography).**



In particular, 25-Nov-2013, 11-Dec-2013 and 27-Dec-2013 show a very similar pattern along all the transects and only 28-Jan-2014 diverges from it, with a smaller difference between maximum and minimum values. However, the main tendency is maintained. This small difference could depend on the different SZA between the scenes. In fact, increasing the SZA, a major heterogeneity was found in both study areas, because of an exponential effect on albedo values, as suggested by Picard et al.

(2016). The first three scenes present a small range of SZA, between 67° and 69°. In contrast, the last scene has a higher SZA of 72°. This discrepancy between scenes with SZA < 70° and > 70° is constant for all the seven analysed transects, which have different wind directions (35°-63°, Fig.2). On average, in the central transect of the Landsat scene (transect C), the albedo of glazed surface is lower by 0.03-0.04 compared to the upwind zone of the dunes, from a NIR albedo range of 0.81-0.84 in the upwind area (snow sastrugi) to a 0.77-0.81 range downwind (glazed surface).

Looking at broadband albedo (Fig.4b), the dominant pattern seen for NIR albedo is still present and again the main intra-annual difference is observed between January and the other scenes. In general, broadband albedo values show lower variability compared to NIR albedo and their range is lower. The highest values, located on the upwind part of the dune, range between 0.78-0.80 while on the downwind areas we found an average albedo of 0.77-0.78. On average, the difference between snow (upwind) and glazed surfaces (downwind) is around 0.02, almost halved compared to NIR albedo. The difference in the

detection ability of NIR and broadband albedo stems from the fact that broadband albedo obtained by using Liang conversion algorithm (Liang, 2001) considers the visible area of the spectrum and the shortwave infrared. Considering the LL transect (Fig.2), from broadband albedo it is hardly possible to recognize the differences between glazed and non-glazed areas, which in the visible wavelengths look very similar (Warren, 1982). Nevertheless, in other parts of the Landsat scene, e.g., LR transect (Fig.2), downwind and upwind areas of the dunes are easily detected, with an albedo difference higher than 0.02.

Focusing on the interannual analysis of the 14 available scenes in the EAIIST area (Table A1), we chose to divide the dataset into two sub-datasets in accordance with the SZA values of each single date, following the reasons previously discussed (Picard et al., 2016). The first sub-dataset considered those scenes with a SZA < 70° and the second one with SZA > 70°.

In Fig.A1, we show four NIR albedo graphs of the scene centre transect (C) considering the 7-scene dataset from 2013-2019. Looking at Fig.A1b-d the differences between the scenes with SZA < 70° and SZA > 70° become evident; in fact, while the

former shows a strong homogeneity among the different images, the latter are more heterogeneous. Above all, NIR albedo on 14-Oct-2015 diverges from all other observations, owing to its very high SZA of 81°. In general, excluding 14-Oct-2015, all the scenes show the same pattern, with higher NIR albedo values in the upwind areas of the dunes and lower albedo in the downwind zones, characterized by the presence of glazed surfaces (Fahnestock et al., 2000; Frezzotti et al., 2002a, b), as already observed in the intra-annual analysis. However, strong differences between different summer seasons are not detected,

and the albedo pattern remains fairly constant.

### 3.2 Brightness temperature

The TOA brightness temperature was here calculated form Band 10 (10.60-11.19 µm) and 11 (11.50-12.51µm) of Landsat 8. Temporally, for both bands we observed an intra-seasonal trend: in fact, while brightness temperature remains ≥ 244K (-29°



C) during the middle of the summer (11-Dec-2013 and 27-Dec-2013), it decreases moving away from the summer solstice.

Temperatures range between 238 and 241 K (-35 / -32° C) on 25-Nov-2013, 26 days from the solstice. The difference increases on the farthest date, 28-Jan-2014 (38 days) with the lowest values ranging between 236 and 239 K (-37 / -34° C). Looking at the 5 transects, we found the same pattern previously observed for the albedo, but proportionally inverse with respect to topography, in both Band 10 (Fig.5a) and 11 (Fig.5b).

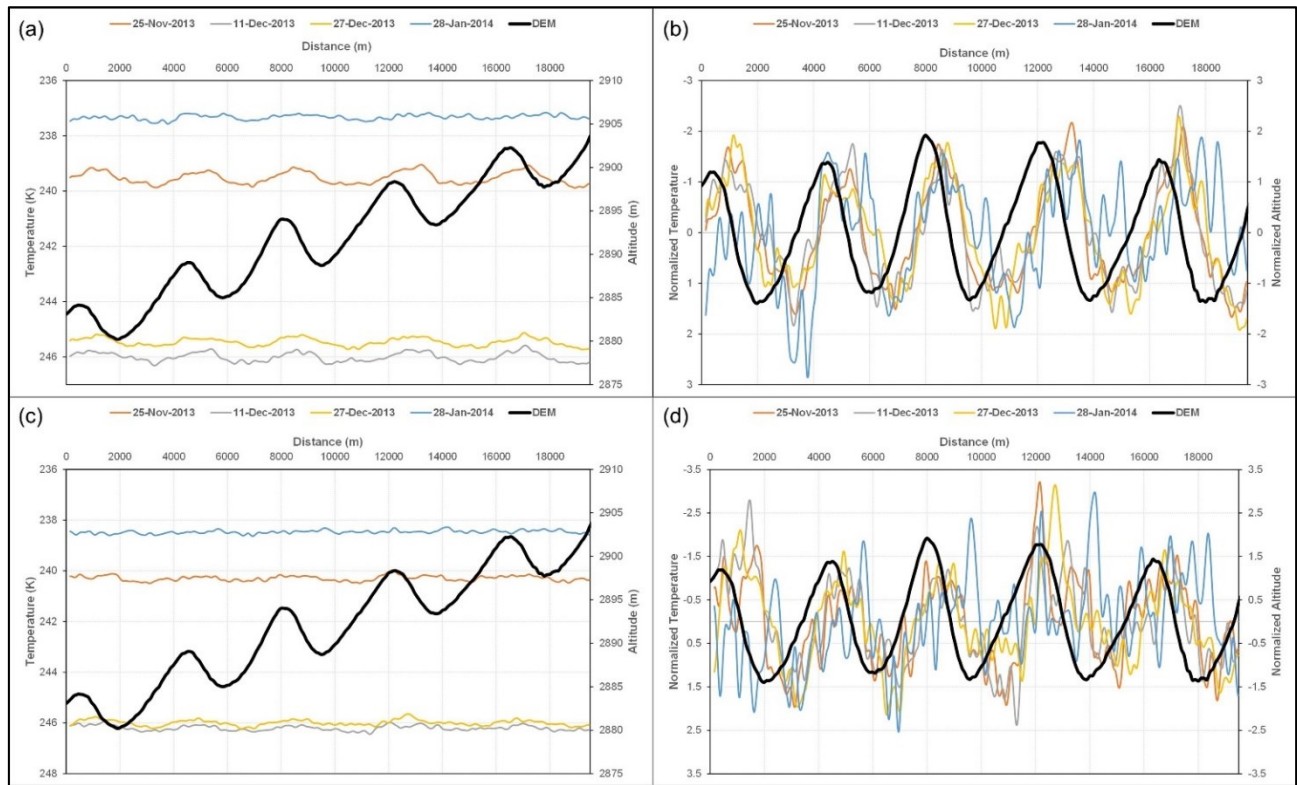

**Figure 5: Thermal infrared Band 10 (a-b) and 11 (c-d) in the austral summer season 2013-2014 for the C transect (see Fig. 2 for location) and elevation from REMA DEM. a-c represent the weighted moving averages based on 11 transect pixels and b-d the normalized moving averages (with detrended topography).**

In fact, higher temperatures correspond to the glazed part of downwind areas of the dunes and conversely, lower values are related to snow sastrugi in the upwind zones, in accordance with Fujii et al. (1987). In general, Band 11 is noisier compared

to Band 10, even if the general pattern can still be detected.

Nevertheless, in comparison with spectral and broadband albedo observations, here the differences between downwind and upwind faces are very small, around 1 K at maximum and are more evident in Landsat 8 OLI Band 10. These differences are directly correlated to the ones observed in albedo, as a higher quantity of energy is absorbed on glazed surfaces. High correlations are also found between NIR albedo and brightness temperature (e.g., R = -0.85 with Band 10 in transect C). Intra-



annually, the difference between glazed surfaces and snow is higher at the end of spring (max 1 K in November) and tends to decrease over time, becoming lower than 0.5 K at the end of summer, where differences between the two surfaces are hardly detectable.

### 3.3 Topographic aspect and slope, wind direction and SPWD

The two study areas, It-ITASE and EAIIST, are in a sloping zone, where, respectively, the altitude increases moving E-W, while the elevation ranges from 2700 to 3200 m in both areas. Thus, the topographic aspect (the direction that a topographic slope faces) is generally E ($\approx 90°$, It-ITASE and $\approx 80°$, EAIIST), including the leeward part of the megadunes with glazed surfaces, and only the windward section faces W (topographic aspect $\approx 270°$, It-ITASE and $\approx 260°$, EAIIST). As regards the topographic slope at a 10 km scale, it is on average 1.5 m km$^{-1}$ and 1.8 m km$^{-1}$ for the It-ITASE and EAIIST areas.

In order to analyse the regional wind direction over the study area, we used ERA5 and sastrugi-based wind direction retrieved from Landsat 8 OLI (Table 2).

| Dataset | Average | Max | Min | Dataset | Average | Max | Min |
|---------|---------|-----|-----|---------|---------|-----|-----|
| Landsat 8 | 224° | 232° | 212° | Landsat 8 | 240° | 250° | 215° |
| ERA5 ≥ 0m/s | 225° | 230° | 220° | ERA5 ≥ 0m/s | 227° | 236° | 215° |
| ERA5 ≥ 3m/s | 225° | 229° | 220° | ERA5 ≥ 3m/s | 226° | 233° | 217° |
| ERA5 ≥ 5m/s | 225° | 229° | 220° | ERA5 ≥ 5m/s | 226° | 234° | 217° |
| ERA5 ≥ 7m/s | 225° | 235° | 220° | ERA5 ≥ 7m/s | 227° | 236° | 218° |
| ERA5 ≥11m/s | 223° | 229° | 216° | ERA5 ≥11m/s | 231° | 240° | 223° |

**Table 2. wind direction statistics (average, maximum and minimum values) for the considered datasets: Landsat 8 at 30 m spatial resolution and ERA5 at 30 km spatial resolution (divided into 5 sub-datasets according to wind speed) at the EAIIST site (left) and It-ITASE (right).**

Concerning the It-ITASE area, ERA5 shows an average meteorological wind direction (direction from where the wind blows) of 227±4°, whereas an average of 240±6° was calculated from the sastrugi. In the EAIIST area, the difference is lower, with a mean of 225±2° from ERA5 and 221±4° based on the sastrugi. The small difference between the two datasets is certainly caused by the fact that the formation of sastrugi is linked to katabatic wind erosion processes (Table 2), and thus represents the direction of high-speed winds. The difference between the average wind direction calculated from ERA5 and Landsat is

rather low and, in fact, becomes even lower (from 4° to 2° and from 13° to 9° in EAIIST and It-ITASE sites respectively) considering a higher wind speed ($\geq 11$ m s$^{-1}$). In addition, this discrepancy could be caused by the different spatial and temporal resolutions between the two datasets (30 m vs 30 km, scene-based vs average of 20 years).

Owing to these small differences, the SPWD is the same when using ERA5 or sastrugi-based wind direction, because the SPWD algorithm considers 8 possible aspect classes. At the regional scale (30 km spatial resolution), the entire megadune

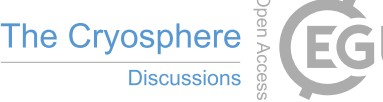



field (almost 500,000 km² area) which covers the two study sites, has an average SPWD of 1.2 m km⁻¹, which is in accordance

with previous studies (e.g., Frezzotti et al., 2002b). To distinguish between leeward (glazed surface) and windward flanks of

the dunes for the two sites (i.e., EAIIST and It-ITASE), the SPWD based on sastrugi was further resampled to 120 m and

thresholds were determined using the 30 sample polygons. For the leeward side of the dunes, a mean SPWD of 5.1±1.2 m km⁻¹ was calculated. In contrast, the windward flanks show negative SPWD values, with a mean of -4.3±1.7 m km⁻¹ (Fig.3a).

**3.4 Correlations between albedo, SPWD and brightness temperature**

From albedo (both broadband and NIR spectral), brightness temperatures and SPWD, we estimated the correlation coefficients

(R) between optical and topographic measures along the transects. Focusing for example on a scene showing one of the lowest

relative SZA (e.g., 27-Dec-2013, EAIIST scene, SZA = 67°), chosen to minimize possible errors in the albedo, we discovered

a high average R for all the transects for each of the considered parameters. Concerning the albedo, NIR α showed higher |R|

values in relationship with the SPWD, averaging -0.60, compared to broadband albedo, whose R decreases to -0.35. Brightness

temperature showed much better statistics, with an average of 0.76 and 0.71, for band 10 and 11 of Landsat 8 OLI, respectively.

In detail, taking into consideration the transect (C), which shows the best statistics, the R for broadband albedo, NIR spectral

albedo and Thermal bands 10-11 was respectively: -0.64, -0.78, 0.87 and 0.77. These high correlation coefficients are easily

observable also looking at the plots that compare the SPWD with albedo and brightness temperature (Fig.6).

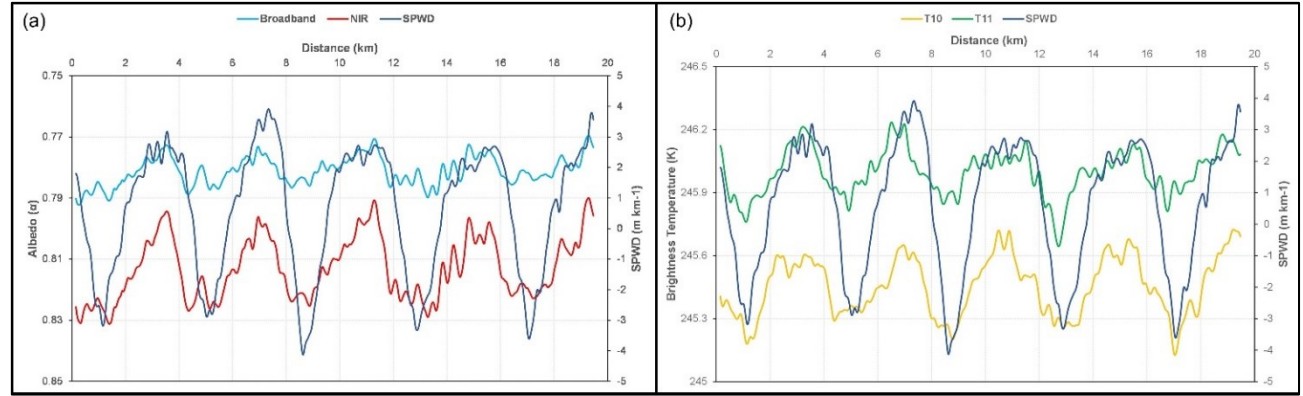


**Figure 6: Transect C plots of SPWD (slope along the prevailing wind direction) at the EAIIST site (Fig. 2 for location), compared to broadband and NIR spectral albedo (a) and brightness temperature from Thermal bands 10 and 11 of Landsat satellite (b) on 27-Dec-2013 (SZA = 67°).**

**3.5 Megadune classification and temporal variability**

By employing SPWD, albedo and brightness temperature, we performed a classification of the megadune field with the aim

of detecting and mapping the downwind (glazed surfaces) and upwind flanks of the dunes. In detail, as NIR spectral albedo

and brightness temperature from band 10 provided the best R statistic with SPWD and showed the highest differences between

the sides of the megadunes, we applied a classification of the areas based on these 3 features. Focusing on an area of the





EAIIST site, which is characterized by wide megadune presence, we provide an intra-annual and inter-annual analysis of
glazed surfaces between 2013 and 2020 (Fig.7).

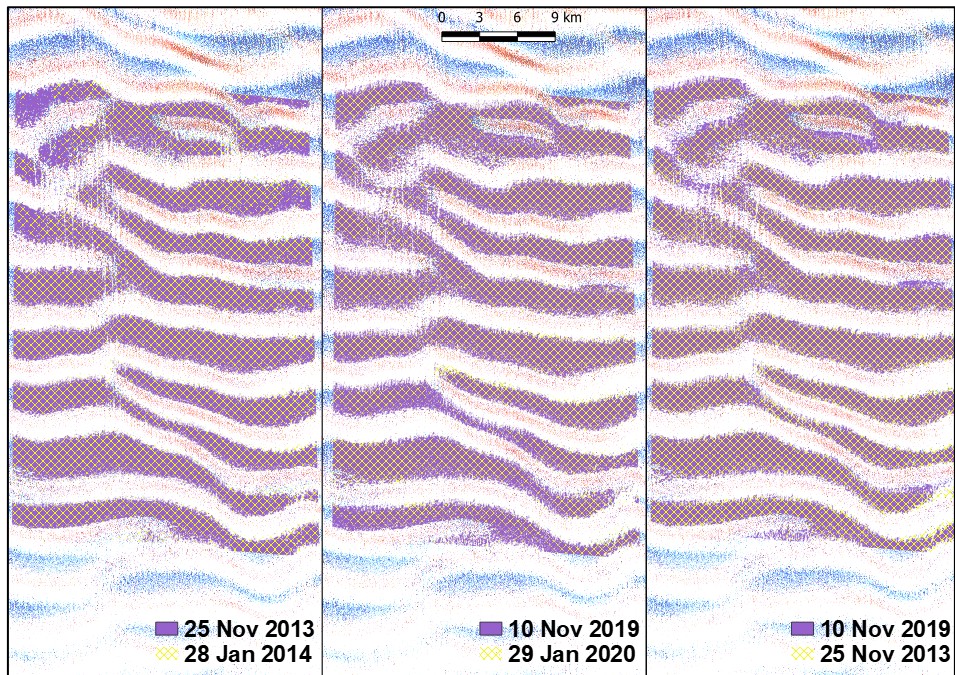

**Figure 7: Wind glazed surface variations in the 2013-2014 and 2019-2020 summer seasons in a sample polygon at the EAIIST site (location in Fig.2a).**

Only two summer seasons (the first one, 2013/14 and the last one, 2019/20) allowed a complete temporal coverage, from the
late spring (November) to the end of summer (January) and in both cases we detected a decrease in the glazed surface between
November and January. In these two summers, the coverage of glazed areas decreased by 16%. Inter-annually, contrasting
results were obtained comparing the dates of January (2014 and 2020) and November (2013 and 2019). In fact, a much higher
increase in area was observed using January scenes, almost 1% (≈0.16% per year) against 0.24% using November scenes
(≈0.04% per year). Fahnestock et al. (2000) pointed out that in late summer, radiative cooling of the uppermost surface layer
leads to the formation of surface frost, by condensation of local atmospheric vapor onto the glazed surface with a more diffuse
specular reflection than in spring, which changes its albedo and does not permit the discrimination between glazed surface and
surrounding snow surface. Considering the It-ITASE and EAIIST sites in those portions included inside the main megadune
field (Fig.1), we have around 20,500 km$^2$ and 33,000 km$^2$ of megadune coverage in the two areas respectively, calculated using
huge megadune polygons by (Fahnestock et al., 2000) (Fig.1). In these areas, approximately 20-25% of the surface is
represented by glazed surface.



### 3.6 Superficial velocity analysis and upwind migration

The calculation of absolute megadune migration was carried out by using feature tracking and by comparing megadune ridges detected on two dates. Regarding the first method, two intra-sensor analyses of Landsat 8 OLI and Sentinel-2 were performed (Table 1). In It-ITASE area, the migration ranges between 11.5±3.8 m $a^{-1}$ (242±12°) for Dec-2016 / Dec-2020 and 14±3.9 m $a^{-1}$ (241±11° direction) for Dec-2014 / Nov-2019. At the other site, EAIIST, a slightly higher heterogeneity was found, as values range between 10.5±4.1 m $a^{-1}$ (219±11°) for Jan-2018 / Jan-2021 and 14.3±3.4 m $a^{-1}$ (225±10°) for Dec-2015 / Dec-2020. The inter-sensor approach using Landsat 7 ETM+ and 8 OLI shows generally lower values (<10 m $a^{-1}$) at both sites, but with an uncertainty estimated by IMCORR higher than the annual velocities, i.e., 21 m $a^{-1}$ on average, which suggests that these results are not statistically robust. In both study areas the migration calculated from Sentinel-2 imagery shows on average a lower velocity compared to Landsat (11.4±3.8 m $a^{-1}$ and 10.5±4.1 m $a^{-1}$ respectively for It-ITASE and EAIIST, Table 1), even if the direction remains similar. At the same time, these results are supported by a larger number of calculated features (an order of magnitude higher), and the higher spatial resolution of Sentinel-2 (10 m) compared to Landsat 8 OLI (30 m). The second method, i.e., megadune ridge vectorization for the same image pairs, shows slightly higher velocities than IMCORR, e.g., 16.7±3 m $a^{-1}$ at the EAIIST site for the period 2013-2019.

The D6 It-ITASE site was covered by 5 GPS-GPR transects along a megadune field; the comparison between GPS elevations (3-Jan-1999) and REMA DEM (02-Feb-2014) provides information about the sedimentological migration of the megadunes during the past 15 years, based on nine megadunes along the traverse. By projecting the transects along the prevalent wind direction (239°), based on the surrounding sastrugi orientation, we were able to evaluate the megadune sedimentological migration between GPS and DEM observations. Considering the crest position of each dune, we calculated the average displacement in this ≈15-year period, obtaining an average superficial speed of 11±5.2 m $a^{-1}$ (Fig.3b). The upstream migration of the dunes is mostly evident on the upwind flank, i.e., accumulation area, in every transect; in contrast, the downwind flank, i.e., ablation part covered by glazed surfaces, remained generally constant in elevation over time (Fig.3b).

## 4 Discussion

### 4.1 Wind and SPWD differences and correlations

At both investigated sites, the direction of the wind from ERA5 at velocity higher than 11 m $s^{-1}$ was found to be closer to the direction of sastrugi surveyed by satellite. This high speed was previously reported by Kodama et al. (1985) and Wendler et al. (1993) to be required for the formation of sastrugi. In fact, calculating the wind direction based on sastrugi orientation, we generally obtained values slightly different from the ones obtained from ERA5 (Hersbach et al., 2020) considering all wind velocity values. Even if the EAIIST site shows similar average directions, i.e., 221° based on sastrugi and 225° from ERA-5, respectively, in the other study area (It-ITASE) a slightly worse agreement is observed, with difference in the average values of 12° (239° for sastrugi-based directions and 227° for ERA-5). However, (Frezzotti et al., 2002b) pointed out that at the it-ITASE site the sastrugi direction (220° – 225°) measured on the field in 1998-1999 is parallel to the sastrugi-glazed surface





field inferred by Landsat 7 ETM+ satellite image recorded in 2000 (230° – 235°). The observed differences can be caused by a number of reasons, including spatial resolution (30 km for ERA 5 compared to 30 m for Landsat). Further still, for
geomorphological reasons the direction retrieved from Landsat is strongly dependent on prevailing winds (katabatic winds), that shape the sastrugi, while ERA5 also takes into account other wind directions than the katabatic. Thus, looking at Table 2, it is evident how higher wind speeds from ERA5, especially ≥ 11 m s$^{-1}$, correspond to wind directions closer to the ones obtained from sastrugi. Nevertheless, the differences between wind direction from ERA5 and sastrugi are low enough to produce an identical SPWD. In fact, with angles in the range 212°-250° (EAIIST minimum and It-ITASE maximum
respectively), all the directions considered in this study, both ERA and sastrugi head towards the SW pixel (202.5°-247.5°) in the SPWD algorithm, with the exception of a few pixels at It-ITASE which show a W direction (angles between 247.5° and 250°). However, with the aim of applying this methodology at large-scale using ERA5 data, e.g., the whole continent, the differences between the two sources can be significant (e.g., at the It-ITASE site), and could produce errors in the SPWD calculation. Therefore, some verifications would be necessary, above all in those areas of Antarctica where forecast
observations are missing.

### 4.2 Roles of albedo, brightness temperature and SPWD to map megadunes and glazed surfaces

Considering that albedo and brightness temperature are not reliable enough for megadune mapping, as unique thresholds cannot be identified to discriminate between glazed surfaces and surrounding snow, the accurate calculation of SPWD is necessary. As showed in Sect. 3.5, it would be possible to detect and properly map glazed surfaces on the leeward flank of
megadunes by combining these three parameters, i.e.: SPWD, NIR albedo and brightness temperature, allowing to study their evolution and trends over time. The SPWD is the only parameter that could be considered as "constant" at 10s km scale, considering the stability of the direction of the katabatic wind, driven mainly by surface slope and the Coriolis force. In contrast, SPWD at km scale, albedo and brightness temperature continuously change annually and during seasons. In fact, NIR albedo significantly varies because of surface changes between the beginning, the middle and end of the summer season (and also
changes though the season in relation to the SZA by ± 0.01-0.02), while brightness temperature varies from a higher temperature near the summer solstice to lower values in late spring and summer, in the range ± 5-10°. In both cases, the differences between leeward (where glazed surfaces are located) and windward flanks of megadunes are not high enough to overcome the seasonal variability and thus a constant range for albedo and temperature is impossible to determine. Therefore, different models in function of the season (beginning, middle and end) would be necessary to properly detect the two sides of
each megadune using automatic methods. However, a near-constant difference between leeward and windward flanks was observed regardless of absolute values. In detail, NIR albedo is on average lower on the leeward flank by 0.04, while broadband albedo by 0.02.



### 4.3 Megadune upwind migration

The MEaSUREs Programme provides ice motion based on synthetic-aperture radar (SAR) interferometry from multiple satellite systems. The SAR image phase centre penetrates up to 10 m on dry and cold firn (Rignot et al., 2001) and provides information on ice flow and not of surface features. In contrast, using feature tracking on optical images (Landsat and Sentinel 2), it is possible to estimate the absolute migration (migration + ice flow) of surface features. The MEaSUREs ice flow velocity at the D6 site (2.2±1.1 m a$^{-1}$) is in agreement with the GPS measurement of 1.46 ±0.04 m a$^{-1}$ (Vittuari et al., 2004). MEaSUREs data shows an ice flow of 6.1±3.4 m a$^{-1}$ (52°) for the EAIIST site. As concerns the absolute migration derived from optical

images, we calculated a weighted velocity based on the number of calculated features for Landsat, obtaining 13.4±3.6 m a$^{-1}$ (It-ITASE) and 12.6±3.5 m a$^{-1}$ (EAIIST). Applying Eq. (5), we obtained a sedimentological migration of 18.6 m a$^{-1}$ (225°) at EAIIST and 15.5 m a$^{-1}$ (239°) at It-ITASE using Landsat 8 OLI data, and 16.5 m a$^{-1}$ (224°) at EAIIST and 13.6 m a$^{-1}$ (239°) at It-ITASE with Sentinel-2. On the other hand, for the transect at the D6 site, we estimated a sedimentological migration of 11 m a$^{-1}$ from the comparison of DEMs from GPS (1999) and REMA (2014). GPS and GPR profiles along the It-ITASE traverse

show the presence of paleo-megadunes buried under the leeward surface of the megadune field (Frezzotti et al., 2002b). Analysis of the D6 firn core allowed to detect the Tambora eruption signal (1816 AD) at 15.36 m depth with an average snow accumulation of 36±1.8 mm we a$^{-1}$ (Frezzotti et al., 2005). Using the isochrone distance of 1.5-1.8 km between the 180 years old paleo-crest and the recent crest from GPS observations (1998-99 AD), we can evaluate the windward migration of the megadune crest at about 8-10 m a$^{-1}$. These field results scalarly summed with an ice flow from GPS of 1.46 ±0.04 m a$^{-1}$ with

a direction of 97° produced an absolute migration of 10.3 m a$^{-1}$ (214°). Therefore, focusing on the It-ITASE site where both remote data and GPS in situ measurements are present, we found good agreement between the satellite and ground measurement datasets, especially with migration calculated from Sentinel-2 images, which in addition shows a significant lower annual error compared to Landsat (1.7 vs 2.6 m a$^{-1}$, Table 1). In fact, the calculated sedimentological migration using Sentinel-2 was 13.6 m a$^{-1}$, very similar to the one obtained from field observations (11 m a$^{-1}$). In addition, the absolute migration

from in situ data (10.3 m a$^{-1}$) is very close to the one calculated using *IMCORR* on Sentinel-2 imagery (11.5 m a$^{-1}$). Hence, we observe a general overestimation of sedimentological and absolute migration using remote-sensing with a mean difference of +1.9 m a$^{-1}$ (uncertainties of 19% for sedimentological migration and 10% for absolute migration). Using Landsat images, larger differences were found, with an average overestimation of 3.8 m a$^{-1}$. This difference could be caused by the fact that with remote sensing a much wider area is included, as opposed to in situ observations which were acquired in transects on a limited

section of the megadune field. Further still, the data were obtained at different time periods. Finally, the spatial resolution and geolocation (Mouginot et al., 2017) could affect the satellite data, as demonstrated in the worse results obtained using Landsat images at 15 m spatial resolution against 10 m of Sentinel. Thus, commercial images of high resolution (≈ 1 m or less) might increase the quality of the calculated results, leading to closer agreement to field observations.

In general, even if the directions between sedimentological migration and ice movement are almost opposite (≈SW and ≈E

respectively), it is evident that the sedimentological process has a more relevant effect on megadune migration, by an order of





magnitude compared to the influence of ice-flow. Sedimentological processes are analogous at It-ITASE and EAIIST. At the second site, a "faster" ice-flow motion was observed, and the velocity of absolute migration is reduced by almost 36%, compared to the initial sedimentological-migration velocity.

Comparison between the elevation profile from It-ITASE measurements and REMA, 15 years apart, clearly shows the upwind
migration of the crest and trough with migration and burying of snow sastrugi over the glazed surface. The glazed surface remains stable in elevation, while the snow-covered surface changes elevation, with a higher accumulation in correspondence with the previous trough, decreasing from the trough towards the windward crest.

## 5 Conclusions

The present study significantly improved the previous knowledge on Antarctic megadunes, confirming previous hypothesis
and providing new relevant information on different aspects of these peculiar landforms. We used recent satellite imagery and applied remote sensing methods in conjunction with field data to investigated in detail two areas on the Antarctic plateau, involved in past international and Italian traverses, respectively EAIIST and It-ITASE. Using the highest resolution and most recent DEM of Antarctica, i.e., REMA DEM, we produced aspect and slope maps of these areas, and combining these data with wind direction, we calculated the slope in the prevailing wind direction (SPWD). In fact, SPWD is a crucial feature in
megadune research and, as pointed out in our results, it has strong correlations with other parameters, i.e. albedo and brightness temperature. In order to obtain wind properties, we used climatic reanalysis data from ERA5, validated against wind direction based on sastrugi, local landforms parallel to katabatic wind flows. Then, starting from Landsat 8 OLI imagery, a first numerical optical analysis of megadunes and in particular of their leeward flanks, covered in glazed surfaces, was carried out, calculating albedo (both broadband and NIR) and brightness temperature. In detail, we discovered that leeward glazed flanks
show a lower albedo (by 0.02) compared to windward snow covered sides; the difference in albedo is even higher (i.e. 0.04) in NIR wavelengths. Albedo and brightness temperature, combined with the SPWD, allowed to produce a detailed and preliminary analysis in time of the extent of glazed surfaces, both intra- and inter-annually. A higher correlation was observed between SPWD and NIR albedo compared to broadband albedo, with an almost doubled correlation coefficient. However, the best agreement is found in the correlation between SPWD and brightness temperature, in particular with Landsat 8 OLI thermal
band 10. We found a general areal reduction of glazed surface through the summer season, with the maximum at the beginning of the summer and the minimum at the end (decrease of -16%). Conversely, inter-annually we observed an areal increase of such surfaces by at maximum 1% in six years. As these features have a near-zero or negative SMB, they deserve major attention and detailed analysis at higher spatial and time scales. Finally, we provided an analysis of megadune migration from field and remote observations. As field data, we used GPS measurements from the It-ITASE traverse and as remote observations and
calculated all the components of megadune migration, i.e., absolute and sedimentological migration and the ice flow. We found good agreement between field and satellite measurements, with an overestimation of absolute migration from satellite data of 10% (which, however, considered a different and more recent period). In fact, from field-based observations we obtained a velocity of 10.3 m a$^{-1}$, compared to 11.5 m a$^{-1}$ with Sentinel-2. Nevertheless, as Landsat 8 OLI showed worse results than
Sentinel-2, having a lower spatial resolution, probably a future implementation of commercial high-resolution satellite imagery

would increase the quality of these results. In this context we can conclude that, since megadunes have an average wavelength of 3 km and migrate approximately at 10 m a$^{-1}$, the burying process of snow on glazed surfaces takes about 300 years and this morphogenetic process decreases the albedo of the dune, NIR and broadband, by 0.04 and 0.02 respectively in such period. In the end, our work points out the importance of sedimentological processes in megadune fields with an "opposite direction" between the migration of surface features and ice flow derived respectively from feature tracking of optical images and SAR.

The reconstruction of paleoclimate based on firn/ice cores drilled in megadune areas or downstream is very complex. In megadune areas, the distortion of recordings is characterized by a snow accumulation/hiatus periodicity of about hundreds of years. The length of periodic variations due to mesoscale relief and/or megadunes depends on ice velocity, megadune migration and snow accumulation, and can therefore vary in space and time. Our results confirm the already hypothesized upwind migration of the windward flanks of megadunes, validating their "antidunal" formation, finding a relative stability in elevation

of their leeward face and upward migration of the crest at ten meters per year.

**Data availability**

Data used to the aims of the present study are available upon request to the corresponding author.

**Author contributions**

 GT, MF conceived the idea of this work. GT and DF developed the procedure and processed the satellite image and data. All

authors contributed to the writing of the final manuscript.

**Competing interests**

 The authors declare that they have no conflict of interest.

**Acknowledgments**

The authors are thankful to MNA–National Antarctic Museum–of Italy (PhD Scholarship of G. Traversa), the Department for

Regional Affairs and Autonomies (DARA) of the Italian Presidency of the Council of Ministers and Levissima Sanpellegrino S.p.A. (post-doc fellowship of D. Fugazza). This study was supported by the EAIIST project (ANR-16-CE01- 0011), the Institut Polaire Français Paul-Emile Victor (IPEV), the National Antarctic Research Program (PNRA), the French Research



National Agency (Project). The authors would like to warmly thank all the participants of the It-ITASE and EAIIST traverses for their tremendous field contributions allowing the collection of the crucial in situ measurements used in this study.

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

**Appendix A**

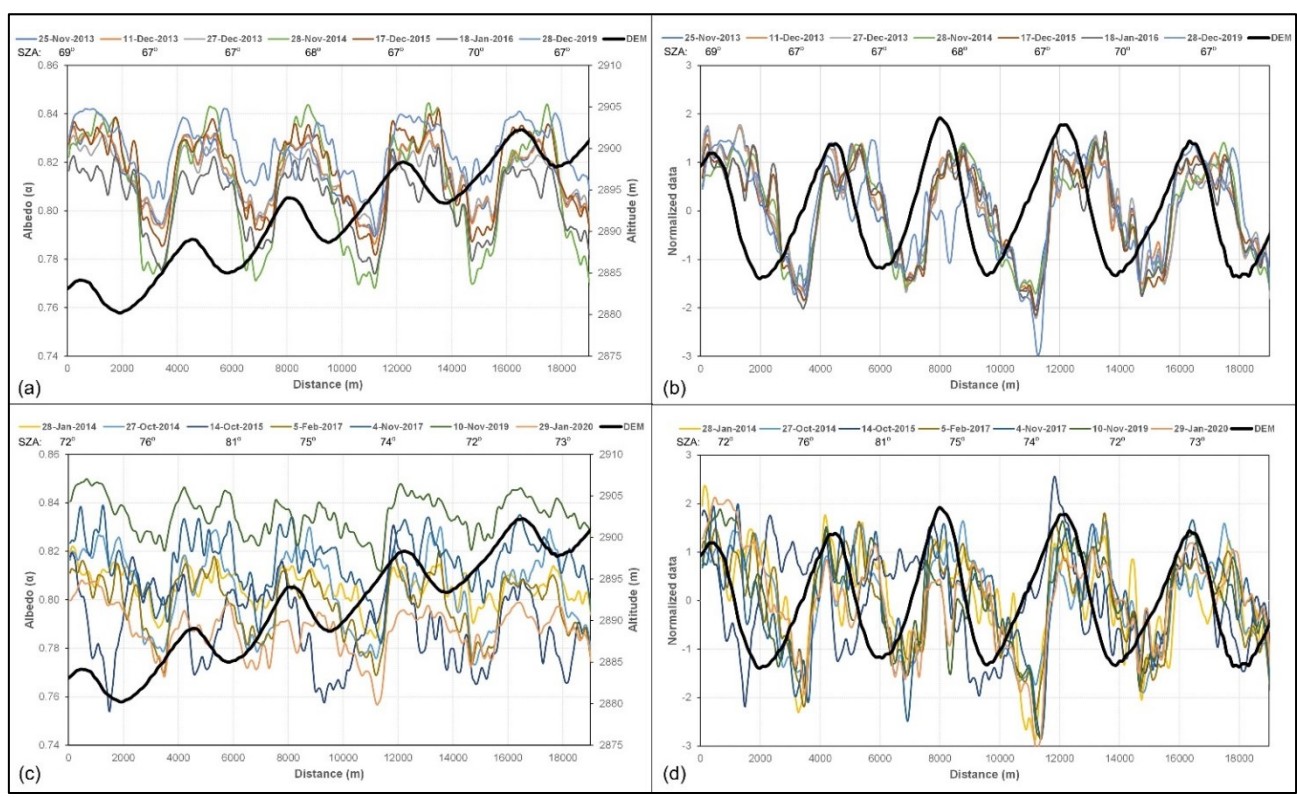

**Figure A1: Transect C NIR albedo at the EAIIST site (Fig. 2 for location) for scenes with a SZA < 70° (a) and > 70° (c) with topography. Normalized NIR albedo of scenes with SZA < 70° (b) and > 70° (d) with detrended topography.**





| Sensor | Tile | Scene | Solar Zenith (deg) | Azimuth (deg) |
|---|---|---|---|---|
| ETM+ | 069119 | 29-Dec-1999 | 67 | 95 |
| ETM+ | 081114 | 02-Jan-2000 | 62 | 71 |
| OLI | 069119 | 25-Nov-2013 | 69 | 89 |
| OLI | 069119 | 11-Dec-2013 | 67 | 91 |
| OLI | 069119 | 27-Dec-2013 | 67 | 93 |
| OLI | 069119 | 28-Jan-2014 | 72 | 95 |
| OLI | 069119 | 27-Oct-2014 | 76 | 87 |
| OLI | 069119 | 28-Nov-2014 | 68 | 89 |
| OLI | 069119 | 14-Oct-2015 | 81 | 87 |
| OLI | 069119 | 17-Dec-2015 | 67 | 92 |
| OLI | 069119 | 18-Jan-2016 | 70 | 95 |
| OLI | 069119 | 05-Feb-2017 | 75 | 95 |
| OLI | 069119 | 04-Nov-2017 | 74 | 87 |
| OLI | 069119 | 10-Nov-2019 | 72 | 88 |
| OLI | 069119 | 28-Dec-2019 | 67 | 93 |
| OLI | 069119 | 29-Jan-2020 | 73 | 95 |
| OLI | 081114 | 31-Oct-2014 | 68 | 62 |
| OLI | 081114 | 02-Dec-2014 | 61 | 65 |
| OLI | 081114 | 18-Dec-2014 | 60 | 67 |
| OLI | 081114 | 06-Jan-2016 | 62 | 69 |
| OLI | 081114 | 21-Sep-2017 | 83 | 61 |
| OLI | 081114 | 30-Nov-2019 | 62 | 65 |
| OLI | 081114 | 17-Jan-2020 | 64 | 70 |
| S2 | T51CWL | 10-Jan-2018 | 67 | 87 |
| S2 | T51CWL | 02-Jan-2021 | 66 | 84 |
| S2 | T52CEA | 13-Dec-2016 | 59 | 59 |
| S2 | T52CEA | 27-Dec-2020 | 59 | 61 |

**Table A1. Landsat (ETM+ and OLI) and Sentinel-2 (S2) images in the EAIIST and It-ITASE areas used in the study with corresponding Solar Zenith and Azimuth angles from the Landsat/Sentinel Metadata.**

| Year | Nº of stripes | Percentage of the total |
|---|---|---|
| 2008 | 5 | 0.4 % |
| 2009 | 13 | 0.9 % |





| 2010 | 27  | 1.9 %  |
|------|-----|--------|
| 2011 | 128 | 9.0 %  |
| 2012 | 27  | 1.9 %  |
| 2013 | 110 | 7.7 %  |
| 2014 | 217 | 15.2 % |
| 2015 | 136 | 9.5 %  |
| 2016 | 593 | 41.6 % |
| 2017 | 169 | 11.9 % |

**Table A2. Frequency of REMA DEM stripes at the EAIIST and It-ITASE sites from different years, based on the REMA strip index.**