# Peer review of "Megadunes in Antarctica: migration and characterization from remote and in situ observations"

_The Cryosphere, 2022_

## Referee Comment (RC2)

[referee-annotated manuscript omitted]

---

## Author Comment (AC2)

**AUTHORS COMMENTS**

The authors thank the Reviewers and Editor for their constructive comments and corrections that have significantly increased the scientific quality of the manuscript and its clarity.
Here we present our answers to the reviewer's comments. In particular the manuscript has been significantly modified and presented more concisely (10% reduction in length), with additional analysis and expanding key points in the discussion section.  In addition, we provide, according to the reviewers' suggestions, detailed comparisons of the data and their correlation along the examined transects, the classification of the glazed surface using topographic, NIR albedo and temperature brightness parameters and clarify the megadune migration processes and implications.
The revised version and a version with tracked changes are provided, but due the manuscript reshaping the tracked change is very difficult to follow.
We hope that the revised version of the manuscript has improved the quality of the text and of the scientific message.

Changes and answers in response to the Reviewer's comments/suggestions (in italic) are highlighted in bold A.

**Reviewer Scambos (Remarks to the Author):**
*GENERAL COMMENT*

*Much as I like this paper, and learned from it, it is not ready for publication. In general, the description of the work is too wordy, to diffuse, and it seems to track the path of the investigation, rather than present the results as they were perceived at the end of the study. The paper could be much shorter, and could move some of the detailed comparisons (e.g. between Landsat and Sentinel2 results, other nuances in the SPWD and GPS work) to supplemental information.*

*I would suggest that the authors lay out a somewhat more direct goal of the research in the abstract and move the statements about confirming past work, setting up the study, to the Introduction. Along the lines of : 'We investigate two EAIS megadune fields with significant past in-situ measurement data, using in addition current imaging sensors (Landsat 8 and Sentinel2), elevation models (REMA), and accumulation models (RACMO) to explore spectral, thermal, and windward slope relationships with a view towards generating a mapping algorithm for time-series investigation. We also use detailed elevation and ice flow data to determine the net migration and the sedimentological migration of the windward face of megadunes. Our study finds strong correlations between ...NIR, ..thermal and ...slope... but with seasonal variations... and a range of accretionary migration rates that imply all or most of the regional accumulation (as determined by RACMO and other models) is gathered in the accretionary faces.' Results indicate. Xxx correlations, and yyy migration rates.... Our study sets a course for more regional evaluations of.....'*

**A: As suggested by both Reviewers and Editor the manuscript has been completely rewritten and presented more concisely (10% reduction in length).**

*I would suggest combining some of the graphics, since there were few material differences between EAIIST and It-ITASE study sites, and not a lot of justification for showing both the absolute and the normalized plots.*

**A: As suggested by the Reviewer, the graphs of figure 4, 5 and 6 have been merged and only an example (Fig. 2) is presented in the manuscript.**

*Also, show the correlation scatter plots for the parameters that correlate highly,, with the correlation line adn statistics.*

**A: A figure with the scatter plot of the parameters and their relationship is now included in figure 3 in the revised manuscript as suggested. We have also expanded the results section on this point, by comparing NIR albedo and SPWD on transect over time and evolution of NIR albedo over each transect. At lines 262-265 of the revised manuscript, we have written: "Along the transects, the correlation of NIR albedo from the different images is high (R2 up to 0.99) during the spring season (24 Nov 2013, 27 Dic 2013) and decreases by the end of the summer and in comparison with the following years, with an R2 of 0.7 only after 2 years (17 Dec 2015) and up to 0.6 after 6 years (Dec 2019). A Similar decrease in correlation occurs from the comparison of the SPWD and NIR albedo from 2013 (R2 0.66) to 2019 (R2 0.39)."**

*There also seems to be a lot of zig-zagging in the text between Methods, Results, and Discussion. Try to iron these out, saying things once and saying the most definite things that you wish to share in each section.*

**A: The manuscript has been largely rewritten and presented more concisely (10% reduction in length).**

**Detailed comments**

*L16 – change to: …taking advantage of the most recent….*
**A: The sentence was removed because the manuscript was shortened according to the referees' suggestions.**

*L19 suggest present tense: analyse, not analysed.*
**A: We have modified the sentence as suggested.**

*L42 – change to: ….that can be observed from satellites…*
**A: The sentence has been removed in the restructuring of the introduction**

*Figure 1 – suggested caption text: Satellite image map of the Antarctic continent (Jezek, 1999) with elevation contour lines at 1000m a.s.l. intervals. Megadune regions are shown as cross-hatched blue areas (Fahnestock et al., 2000), with net surface mass balance in color for areas with SMB < 50 kg m-2 yr-1, based on RACMO (van Wessem et al., 2014). The main study areas, shown as dark boxes in panel a, are the EAIIST region (panel b) and It-ITASE sites, both represented by Landsat 8 images (give path, row, and dates of Landsat acquisitions). (no need to mention the projection)*
**A: The caption text was partially modified as suggested, except for some additional modifications due to manuscript restructuring. Also, Figure 1 and 2 were merged together. The caption now reads: "Figure 1: Location map of megadune area: (a) Satellite image map of the Antarctic continent (Jezek, 1999) with elevation contour lines at 1000 m a.s.l. intervals, megadune regions are shown as cross-hatched blue areas (Fahnestock et al., 2000), with snow precipitation by RAMCO in colour for areas with precipitation < 50 kg m2 a-1 (Van Wessem et al., 2014), black rectangle (b box). (b) The megadune field with two study sites, EAIIST red rectangle (c box) and It-ITASE blue rectangle (d box). (c) Landsat 8 OLI image in false colour (069119 scene, 17/Dec/2015) of the EAIIST area; the red polygon is the area for the analysis of variations of glazed surfaces (Fig. 4). (d) Landsat 8 OLI image in false colour (081114 on 18/Dec/2014) of the It-ITASE area and D6 core site; the green rectangle shows the location of Fig. 5. In (c) and (d) boxes, red arrows represent ERA5 wind direction and green arrows sastrugi-based wind direction, while the yellow lines show the location of the transects studied."**

*L63 – change to ….provide a detailed survey of Antarctica's megadunes using remote sensing….*
**A: We have modified the sentence as "The aim of the study is to provide a detailed survey of two megadune areas using remote sensing data"**

*L65 – suggested notation, ….two Landsat 8 scenes, P069 R119 (EAIIST site) and P081 R114 (It-ITASE site), in order…. I suggest not giving a lat-long point here, the scenes cover a large area. Perhaps you could show the corner positions of the scenes in Figure 1, outside the images at their corners.*
**A: We have rephrased the sentence to focus on the areas and have added "centered at" for lat-long. We have also added the coordinates of scenes in figure 1.**

*L67 – in what band will you provide brightness temperature – thermal? passive microwave? (thermal, ok).*
**A: Yes, it is thermal TRS1. This is now specified throughout the manuscript.**

*L90-91 – This sentence is a bit odd – the katabatic wind direction is known from models and wind observations; this wind direction is fundamental to megadune orientation, not the reverse. I think this sentence could be removed or converted to a different sentence about the katabatic winds and megadunes.*
***A: Thank you for pointing this out. The sentence has been removed in the shortening of the manuscript***

*L91-92-93; the SPWD slopes of the leeward and windward sides of megadunes are of opposite sign -- please note that.*
**A: The sentence has been removed owing to the restructuring of the manuscript; we now point out the differences in sign of the SPWD in the results section 3.1, at lines 309-311, where we have written: " For the SPWD on megadunes, we found a mean value of 5.6±1.0 m km-1 for the leeward side and negative SPWD values, with a mean of -4.2±1.6 m km-1  on the windward flanks."**

*L126 – no need to capitalize 'metadata'.*
**A: We have removed the capitalization throughout the manuscript**

*L165 – these sentences, beginning with 'The transect plots…' are hard to understand, I suggest rewording them and referring to Figure 2 and perhaps other figures.*
**A: This information has been moved to section 2.2.1, at lines 205-208 of the revised manuscript, where we have written: "Moreover, we determined the strength of the relationship between SPWD vs NIR albedo, and SPWD vs thermal brightness temperature (applied on the moving averages of 11 pixels weighted based on the distance from the central point) using linear regression. The comparison analysis was conducted at seasonal scale for the 2013-2014 (4 images) and at pluriannual scale on 17 images distributed over 8 years."**
**Additionally, the information is reported in the caption of Figure 2 in the revised manuscript which now reads: "Figure 2: (a) moving average (based on 11 transect pixels) of NIR albedo (α) between November 2013 and February 2014 for transect C at the EAIIST site (see Fig. 1c for location) and elevation from REMA DEM. Corresponding normalised moving average of NIR albedo (b) and thermal brightness temperature TIRS1 (c) during the austral summer season 2013-2014 for transect C and elevation from REMA DEM (detrended topography)."**

*Figure 2 caption – add 'lines' : …of transects (yellow lines) on the …. No need for the latlong positions, the images cover extensive areas. …green rectangle (b) is the area shown in Figure 3a…. NOTE: if you did not reproject the L8 images, they are in polar stereographic projection, not UTM – that is how they are distributed (all images south of 60°S latitude are in polar stereographic).*
**A: Figure 2 has now been merged with Figure 1 and the caption has been modified accordingly. See comment on Figure 1 for details on the new caption.**

*L179 'firstly' is not wrong, but old-fashioned – suggest change to 'first'*
**A: We have replaced "firstly" with "first" as suggested**

*L180-185 FYI, the USGS is now providing 'analysis ready data' which in fact includes TOA reflectance. I am not certain that this extends globally yet, but a request to USGS to specially process a handful of images would be worth trying. This can be outside this paper, **but** if the 'analysis ready data' is available, it should compared with your work.*
**A:  Thank you for the comment. Unfortunately, the data are not available for the Antarctic continent; it would be definitely worth checking them in future work if they become available**

*L198-204 This could be a significant issue: the Landsat 8 thermal channels 10 and 11 had some problems, and in fact it was recommended that channel 11 not be used for analysis. Depending upon when you retrieved L8 data, it may or may not have had a corrected channel 10 value, corrected for stray light impacts and pushbroom detector noise.*
**A: We now only consider band 10 TRS1 in the analysis and have checked that band 10 had stray light correction implemented**

*L214-216 section between the commas: … , where Band 5 NIR …. Frezzotti et al., 2002b), … Remove from the sentence, perhaps find another place for these words. It is distracting from your edge-detection of sastrugi method for determining wind direction from the imagery.*
**A: We have removed this sentence as part of the restructuring of the manuscript.**

*L218 – what was the variation in degrees between (a) extracted wind directions in a uniform section of the images, and (b) among the wind directions determined in the repeated imagery for the same areas? 'Only small differences'… I'm sure you are right, but a value in degrees would be useful to underscore that.*
**A: The differences are smaller than 5° in both cases. We have now added this information in the manuscript at lines 300-302, where we have written: "The analysis of sastrugi direction using 7 Landsat scenes from the spring and summer months during the period 2013-2020 show small differences in direction within each image and in repeated imagery (< 5°), confirming the stability in direction of sastrugi landforms and thus the persistence of katabatic wind. "**

*L222-230 – I suspect this adjacent-pixel method was a bit noisy – and you are saying that determining the SPWD over a 90m cell (3x3 pixels) would significantly reduce the slope? This does not seem right to me. Also – I'm not seeing how Equation 4 does not include either trigonometric functions, or, more than two pixels with some kind of ratio for the elevations of the windward pixels - ?*
**A: We have recalculated the SPWD using the method described by Scambos et al. (2012), with wind direction derived from ERA5 and sastrugi. The differences between the two methods are rather small, i.e. 1 m km-1. We nevertheless chose to use the methods by Scambos et al. (2012). We have therefore replaced the text at lines 197-201 of the revised manuscript with: "To further estimate the SPWD based on the wind direction from ERA5 and Landsat-derived sastrugi, we used the approach of Scambos et al. (2012), i.e., we calculated the dot product between the slope derived from the REMA DEM and the wind direction. The algorithm was applied to ERA5 and sastrugi-based wind directions resampled at 120 m spatial resolution, and the REMA DEM was resampled to match ERA5 and sastrugi-based wind directions using bilinear interpolation. The resulting SPWD has units of m km$^{-1}$."**
**All the SPWD values mentioned in the revised manuscript now reflect the changes in the method used to calculate it.**

*L233 'modules' is not the right word here – 'modes' might be what you mean, but while it sort of works, the meaning is unclear. Perhaps just end the sentence at 'directions'.*
**A: We have replaced "modules" with "intensity" here.**

*L234 remove 'thus', not needed.*
**A: We have removed "thus" as suggested**

*Figure 3. It would be a bit better to flip the x-axis of 3b around, since 3a shows the wind moving from left to right, and the topography goes downhill left to right as well.*
**A: We have remade figure 3 (figure 5 in the revised manuscript) by adding more GPS survey transects acquired at It-ITASE and flipped the x axis around as suggested and add the internal layering from GPR.**

*L250 This approach may have a problem. In Landsat 8, there is a strong correlation, even spanning years, to the linear sastrugi pattern and 'surface roughness' at the decameter scale; the megadunes themselves are much 'softer' features and are probably not the features that would be tracked by IMCORR (or PyCORR – see GoLIVE data at NSIDC; or ITS_LIVE data at NSIDC as well). You could address this by filtering --- use a high-pass filter of ~150m length scale on the image pairs to isolate the sastrugi pattern and erase the megadunes; and a low-pass filter of the same scale to smooth out the sastrugi and leave the megadune features for IMCORR or PyCORR. You may want to use a large highpass filter as well for the megadunes (~6km), to supress bedrock-driven features (the 'undulation field') from the megadunes-only image pair. Note you would need to use a large reference area size to track the megadune pattern after filtering (or downsample the images, or both). This should allow a direct comparison of the two motion maps you are after. The high-pass filtered mapping should isolate the true ice sheet flow, directly downhill; and the low-pass filtered map should emphasize the megadune migration, a combination of sedimentological advance and ice flow. I see in L270 you attempted this with edge-detection of glaze-accumulation zones.*
**A: We have applied a Fast Fourier Transform with the suggested wavelengths before performing megadune feature tracking and sastrugi identification. We have added a description of this process at lines 234-235 of the revised manuscript, where we have written: "Prior to feature tracking, each image pair was pre-processed by using a low pass filter with a length scale of 150 m implemented through a Fast Fourier Transform to smooth out the sastrugi and leave megadune features for tracking", and at lines 193-194 of the manuscript, where we have written: " Prior to edge detection, each image was pre-processed by using a high pass filter with a length scale of 150 m implemented through a Fast Fourier Transform to highlight the sastrugi." In the revised manuscript, we report the new values of megadune movement and sastrugi-based wind speed after this operation. Differences between the two methods are in the order of 1%.**

*L257-259 – please include these attempts in the table.*
**A: The results from the comparison of Landsat 8 and 7 ETM+ are unreliable; therefore, we decided to remove them from the manuscript altogether.**

*L290-309 – I think this section could be stated more briefly and simply. Also – did you explore surface grain size or a normalized red-infrared band difference? NDSI?*

**A: Thank you for the comment. We have considerably shortened the section. We did not use other approaches but plan to do so in future work. We have therefore added a sentence in the conclusion section pointing at future developments, at lines 477-478 of the revised manuscript, which reads: "Further research might consider other parameters to automatically detect snow glazed surfaces, including snow grain size or the normalised difference snow index."**

*L295 and Figure A1 – do you have any explanation for the decreased albedo with increased SZA? (sastrugi shadowing…).*
**A: Please note that we have now removed images with a SZA higher than 75°, as suggested by the other referee. Sastrugi shadowing is probably the driver of higher NIR albedo variability of windward flank. We have therefore added a sentence in the discussion section, at lines 399-401 of the revised manuscript, which reads: "The observed change on NIR albedo and brightness temperature on the windward flank is correlated to the sastrugi formation and deterioration during the season, and their relative change in shadow (Warren, 1982)."**

*L322-328 – again please check – it may be that for this analysis, brightness temperature only, and perhaps with some spatial averaging of values? This application will be ok – but prior to 2020 there were significant issues with Band 11 in Landsat 8 (which were partiall addressed by processing for the entire archive in 2020). It would be better to base your brightness temperature solely on Band10*
**A: We now consider only Landsat 8 Band 10 for calculation of brightness temperatures as suggested in your other comment and have modified the manuscript accordingly throughout.**

*Figure 5 Please re-plot with the y-axis warmer=up!*
**A: We have merged figures 4 and 5 in one, showing normalized temperature and NIR albedo, now shown in Figure 2 in the revised manuscript. We chose to maintain the y axis with the warmer temperatures down to ease comparison with the megadune topography, as temperatures and topography show an inverse relationship.**

*L347 change to '….is generally east (….*
**A: We have modified the sentence accordingly.**

*L348-349 change to '…The regional topographic slope (10 km scale) is on average 1.5 m km-1….'*
**A: We have modified the sentence accordingly.**

*Table2 – please put the regions in the upper left of each sub-table, EAIIST (left) and It-ITASE(right).*
**A: We have modified the table as suggested**

*L355-362 this could be written more concisely.*
**A: We have restructured the paragraph to shorten it as suggested. At lines 303-305 in the revised manuscript we have written: "The comparison of the results of wind direction obtained using sastrugi direction by satellite (resampled using bilinear interpolation) and ERA5 present similar values for both areas, with lower difference in the EAIIST area (differences of 1° in average values) compared to It-ITASE (9-14°)."**

*Section 3.4 – the goal of this section could be presented more concisely with crossplots of*

*the parameters showing the strong correlations.*

*Overall - -most of this up to this point is nice to see, but not a surprise – NIR albedo lower, temperature higher, SPWD trends, sastrugi versus model wind, all these are tightly correlated and are a function of the published characteristics and formation ideas for megadunes. So, while I understand that it was work to put it all together, and you'd like to show it, it is much more interesting that you combined them to create a classification method for megadunes that you can use to look for seasonal and interannual changes. Section 3.5 – I think this is the best part of the manuscript – a slghtly quicker pace to get to this part of the paper might be better.*

**A: We have considerably shortened the manuscript to get more quickly to the megadune mapping section. Section 3.1 now reports all the information on the evolution of NIR albedo and thermal brightness temperature on the transects and the megadune mapping and their relationship. We now show in section 3.1 a new figure (figure 3 in the revised manuscript) with crossplots of NIR albedo and brightness temperature against the SPWD as suggested.**

*Figure 7 – can you present this as an image with the change (glaze in November not January; and glaze in January not November) shown as colored strips on the black-andwhite NIR image?*

**A: Unfortunately, as the REMA DEM is only available from one date and so is the SPWD which is based on it, and owing to the megadune migration, applying an interannual classification based on albedo, brightness temperature and SPWD would be unreliable. Therefore, we have removed this analysis and now perform a comparison of the detection with/without SPWD and by applying thresholds for these variables on an entire Landsat tile and a narrow area. At lines 312-317 of the revised manuscript, we have written: "Applying the automatic detection on the entire Landsat scene from 17-Dec-2015, when excluding the SPWD, approximately 34% of the entire tile was detected as glazed snow, compared to 24% using also SPWD. On the smaller area instead, a slight variation was detected with and without SPWD (22% and 23% respectively). Clipping the glazed snow surface estimated on the entire Landsat 8 tile by using the tile-based thresholds to the extent of the narrower area, an overestimation of 70% was found in comparison with the results obtained directly on the smaller area with the area-specific thresholds and when using the SPWD, rising to +88% without SPWD (Fig. 4)."**

**We also now report in section 3.1 correlations between NIR albedo and SPWD on transects on different dates to show that NIR albedo and SPWD decorrelate over time and point out that the SPWD from a different date would be needed for the classification. At lines 262-265 of the revised manuscript, we have written: "Along the transects, the correlation of NIR albedo from the different images is high (R2 up to 0.99) during the spring season (24 Nov 2013, 27 Dic 2013) and decreases by the end of the summer and in comparison with the following years, with an R2 of 0.7 only after 2 years (17 Dec 2015) and up to 0.6 after 6 years (Dec 2019). A Similar decrease in correlation occurs from the comparison of the SPWD and NIR albedo from 2013 (R2 0.66) to 2019 (R2 0.39)."**

**The suggestions for the coloring scheme have been followed in the new image (Figure 5 in the revised manuscript).**

*L405 – Can you assemble Landsat images of the entire dune area for, e.g. 2013 and 2020, and look for regional expansion of glaze areas in January? This would be a very important result.*

**A: We have expanded the detection to the entire Landsat tile; however, unfortunately we cannot provide a reliable estimate of the interannual variability, as explained in detail in the comment above.**

*Section 3.6 – 'Superficial' in English means 'unimportant' or 'trivial'– I think just 'Ice sheet velocity and upwind megadune migration' would make a better heading here.*
**A: We have replaced the title with "megadune migration".**

*L422 – do you have a figure of the nine megadunes traversed by GPS? I see Figure 3b, but perhaps a graphic highlighting the GPS plus REMA assessment of migration?*
*Also – what local accumulation rates are indicated by the frontal accretion? Assuming the glaze areas on the lee side of the dunes have near-zero accumulation, what does this mean for the regional accumulation rate, e.g. from RACMO, compared to what you observe?*
**A: Figure 5 in the revised manuscript now shows more GPS transects with REMA and GPS elevation, and also GPR layering to clarify the megadune migration. We have added a discussion on accumulation rates at lines 415-422 of the revised manuscript, where we have written: "The elevation change during 15 years observed using GPS and REMA shows a relative increase of accumulation on the windward flank with the maximum value at the trough compared to the glazed surface area from 29 to 46 mm w.e. a-1 with an average value of 34 mm w.e. a-1, using a density of 360 kg m3 in the first two metres. This value is very close to the estimated change of accumulation in the megadune area from 7 to 35 mm w.e. a-1 provided by Frezzotti et al., (2002b) using the variability of GPR internal layering at the megadune site (Fig. 5). The minimum value represents a decrease in accumulation up to 75% or more on glazed surfaces. The relative stability of glazed surfaces with respect to elevation change and NIR albedo confirms the extremely stable SMB low value of the glazed surfaces with respect to accumulation area, due to the long-term hiatus in SMB forced by wind scouring processes. "**

*L444 – change to 'near-identical' or 'identical within the limits of determination'*
**A: The sentence has been removed in the restructuring of the manuscript.**

*L451-467 – could you not evaluate the inter-annual changes using only, e.g. mid-January images?*
**A: We have removed the inter-annual analysis, as it is not possible to perform this with the SPWD from a single date, as explained in detail in the comment to Figure 7.**

*L484 – 'scalarly summed?' would it not be a vector sum to get the net migration?*
**A: We have removed "scalarly" here.**

---

## Author Comment (AC3)

**AUTHORS COMMENTS**

The authors thank the Reviewers and Editor for their constructive comments and corrections that have significantly increased the scientific quality of the manuscript and its clarity.
Here we present our answers to the reviewer's comments. In particular the manuscript has been significantly modified and presented more concisely (10% reduction in length), with additional analysis and expanding key points in the discussion section.  In addition, we provide, according to the reviewers' suggestions, detailed comparisons of the data and their correlation along the examined transects, the classification of the glazed surface using topographic, NIR albedo and temperature brightness parameters and clarify the megadune migration processes and implications.
The revised version and a version with tracked changes are provided, but due the manuscript reshaping the tracked change is very difficult to follow.
We hope that the revised version of the manuscript has improved the quality of the text and of the scientific message.

Changes and answers in response to the Reviewer's comments/suggestions (in italic) are highlighted in bold A.

**Reviewer Lhermitte (Remarks to the Author):**

MAJOR COMMENTS
*Although the paper tackles an interesting research topic (assessing spatial variations in megadunes) with novel results (upward migration and role on SMB), it may eventually warrant publication if some very major comments are addressed. The major comments are mostly related to a complete reorganization of the paper, which would require a significant effort. The major comments are outlined below and identified in detail in the specific comments are made in the uploaded pdf.*
*• The paper is currently written in a very lengthy and narrative setup following the research path with parts of the data, methods and results diluted throughout the paper. This makes it difficult to quickly read the paper and/or look for specific data set processing, analyses, etc. Reorganizing this into better aligned data, methods and results sections will allow to shorten and focus the paper better highlighting the main message.*

**A: As suggested by both Reviewers and Editor the manuscript has been completely rewritten and presented more concisely (10% reduction in length).**

*• The paper shows some direct overlap with a previous conference proceeding by the same authors ( Traversa, G., Fugazza, D., and Frezzotti, M.: Analysis of Megadune Fields in Antarctica, in: 2021 IEEE International Geoscience and Remote Sensing Symposium IGARSS, 5513–5516, https://doi.org/10.1109/IGARSS47720.2021.9554827, 2021a). I would consider removing the overlap (e.g. again showing the NIR profiles) and focusing on the novelty in this paper*

**A: The overlap with the previous paper has been removed and the manuscript now focuses more on the novelty of the present work; the analysis of broadband albedo is not discussed in the present paper, as it was in the Traversa 2021 paper. The present manuscript is focused on the analysis of SPWD, NIR Albedo and brightness temperature with the aim to create a method for the automatic detection of snow glazed surfaces of megadunes and on the detailed quantification of their migration.**

*• The migration problem remains complex as the Traversa et al show that the windward flanks migrate, while the leeward flanks don't. Consequently, it cannot be a (moving) steady state and after a long time the windward flanks would overlap with the leeward flanks, which would seem rather problematic on the long term. Therefore, the migration part of the study would benefit from some extension to address/document the importance of this discrepancy in more detail.*

**A: The quantification of megadune migration is one of the main results of the manuscript and is now significantly improved and discussed implication for morphologies overlap and their implication for ice core drilled in the megadune area. We have expanded the text in the results section 3.1 and added comments in the discussion and conclusion section. In the discussion section, at lines 430-433 we have written: "The results allowed us to calculate all the components of migration and to conclude that for a megadune with a wavelength of 3 km we could calculate an absolute migration of approximately 10 m a$^{-1}$. This burying process of snow on glazed surfaces takes about 300 years, with overlap of crest to through and glazed to sastrugi surface as observed by GPR internal layering (Fig. 5). ". In the conclusion section, at lines 480-483, we have written: "The results obtained using field measurements and remote observations allow to calculate all the components of megadune migration, absolute (11-14 m a$^{-1}$),**

sedimentological migration (13-15 m a$^{-1}$) and the ice flow (1-2 m a$^{-1}$) and to conclude that for a megadunes with a wavelength of 3 km and migration of approximately 10 m a$^{-1}$, the burying process of snow on glazed surfaces takes about 300 years, with overlap of crest to through and glazed to sastrugi surface." We have also added additional GPS transects in Figure 5 and GPR layering to better show the process.

• *The introduction now contains two separate parts with a general introduction with short summary, focus, aim and is then followed by a second more in-depth introduction about the processes, uncertainties related to megadunes. I would advice to go for general introduction -> in-depth introduction (including scientific problem statement) -> aim -> short summary. This will increase the readability and flow of the introduction.*

**A: As suggested by the Reviewer, the Introduction was completely rewritten as proposed.**

• *The subtle differences in use of different data sets (Landsat, Sentinel-1) and preprocessing (e.g. FCC on different bands for different data sets) makes it complex to follow the flow and setup of the paper as it therefore reads as a patchwork of different things. Consider switching to a more homogeneous or grouped approach that would allow the reader to better understand (Study Area / Data including processing (Landsat, Sentinel, Wind, GPR, Velocity)) / Methods (reflectance + albedo, thermal brightness temperature, SPWD, classification, migration (including comparison with existing velocity) / Results per method subsection / Discussion without new results / Conclusion ).*

**A: Following the Reviewer's comments, the manuscript has been reorganized with the following sections: 1.0) introduction, 1.1 Study area, 2 Data and Methods, 2.1 Data, 2.1.1 Satellite datasets, 2.1.2 Atmospheric reanalysis dataset, 2.1.3 Topographic dataset (DEM), 2.2 Methods, 2.2.1 Automatic detection of glazed snow surfaces, 2.2.2 Megadune movement estimation, 3 Results, 3.1 Megadune characterization and automatic detection, 3.2 Megadune migration, 4 Discussion, 4.1 Application of the automatic detection of glazed snow on megadune fields , 4.2 Megadune upwind migration, 5 Conclusions**

• *The paper mentions seven transects for the analysis, but from my understanding many of analysis seem limited to one transect (C in figures 4-6). Consider making the analysis more general and extensive so the results can also be generalised for the other transects.*
*A: On the basis of the comments from Reviewer 1, we present in figure 2 of the revised manuscript only transect C; in figure 3 we show instead scatterplots considering all transects. The analysis and description from the results section 3.1 and discussion section 4.1 take in consideration all the transects analyzed.*

• *Based on the equation of SPWD (Eq.4) the SPWD is only calculated for eight potential neighboring cell (i.e. in steps of 45 degrees). This implies that wind uncertainties of 22.5 degrees (and corresponding height differences in different directions) would not affect the affect the SPWD as the wind can only flow N,NE,E,SE,S,SW,W,NE and nowhere in between. This could have large impact on the SPWD as the 8 directions do not necessarily align with the maximal slope alogn the terrain. I would therefore recommend to recalculate the SPWD along the real wind direction but for interpolated DEM data.*
**A: The SPWD has now been recalculated based on the method proposed by Scambos et al. (2012). While differences between the previous method and the new one are small, i.e. 0.1 m**

km$^{-1}$. We chose to use the method by Scambos et al. (2012) as it was applied before for the same purpose. At lines 197-201 in the revised manuscript, we have written:

"To further estimate the SPWD based on the wind direction from ERA5 and Landsat-derived sastrugi, we used the approach of Scambos et al. (2012), i.e., we calculated the dot product between the slope derived from the REMA DEM and the wind direction. The algorithm was applied to ERA5 and sastrugi-based wind directions resampled at 120 m spatial resolution, and the REMA DEM was resampled to match ERA5 and sastrugi-based wind directions using bilinear interpolation. The resulting SPWD has units of m km$^{-1}$.".
 The values of SPWD elsewhere in the manuscript reflect these changes in the method.

• *Given the difference in satellite response for images with SZA < 70 and >70 I would consider only using images with SZA<70. The other seem erroneous and by limiting it to SZA<70 (in the data section) based on know artefacts (e.g. Picard 2016) it would allow the story to focus on the main points.*
A: We have now removed scenes with SZA > 75°

• *The classification of Figure 7 is completely unclear as all methodological details are missing. Additionally I am missing analyses that show that albedo and brightness temperature cannot be used and/or that the method works and is reliable. Just using a method and saying that it works is not how it should be done.*
A: The sentences suggesting that albedo and temperatures cannot be used have been removed. The methodological details on the classification have now been added at lines 209-219 of the revised manuscript, where we have written: "After having visually identified the thresholds of SPWD, NIR albedo and thermal brightness temperature on the image from 17/Dec/2015, which was one of the best available images in terms of cloud cover (~0%) and presenting no blowing snow/fog and the lowest SZA (67°) for the EAIIST site, we applied a conditional calculation to automatically map glazed snow. In order to better define the detection method and evaluate the use of each parameter, we first applied the automatic detection on the entire tile of Landsat (path 69, row 119; ~36,000 km$^2$) and then on a narrower area (~2,400 km$^2$), which showed higher homogeneity in NIR and TIRS 1 bands. Different thresholds of NIR albedo and thermal brightness temperature were used, which allowed the detection of the highest amount of glazed snow surface in the entire analysed area. The thresholds were defined as follows: NIR albedo < 0.82 and thermal brightness temperature > 246.5 K on the entire tile and NIR albedo < 0.805 and thermal brightness temperature > 247.6 K on the narrower area. Additionally, to determine the role of the SPWD in the automatic detection, we performed the classification by first excluding and then including this parameter (SPWD > 1 m km$^{-1}$)."

• *I would advice to deposit the data corresponding to the paper in a open access repository (with doi) and not rely on requests to the corresponding author.*
A: We have added details on the repositories where data used in this paper are available.

**Comments/Remarks from pdf annotated manuscript**

*L15-17 - Can be removed. Distracts from the story in the abstract*
A:We have removed the sentence as suggested.

*L17-18 - The authors often start sentences with subordinate clauses like "Focusing in" or "Considering" "As regards/Regarding" and I think it removes the flow from the sentence by putting*

*accent on the side ideas. We analysed the dynamic parameters of megadunes, their albedo and morphology for two sample areas ...*

**A: We have restructured and shortened the manuscript as suggested by you and the other reviewer and avoid using subordinate clauses at the start of sentences whenever possible.**

*L20 - I am not sure the meaning of sedimentological would be 100% clear for the reader in this context. Perhaps clarify*

***A: We have added: "i.e. given by the deposition of snow" to clarify***

*L21-22 - If the windward flanks migrate, while the leeward flanks don't, it cannot be a moving steady state and after a long time the windward flanks would overlap with the leeward flanks, which would seem rather unlogical.*

**A: We have added comments in the discussion and conclusion sections, at lines 430-433 and 480-483 of the revised manuscript, to better explain the megadune migration process. See major comment for the full text.**

*L25-26 - Why would you need to do a detailed mapping? Is mapping what is really interesting here?*

**A: The aims of our work are to identify parameters for the automatic detection of glazed surfaces of megadunes and to quantify their migration. We now make this clearer in the introduction section at lines 72-79 of the revised manuscript, where we have written: "The aim of the study is to provide a detailed survey of the spatial and temporal variability of two megadune areas using remote sensing data (Landsat 8 and Sentinel-2), elevation models (Reference Elevation Model of Antarctica REMA, Howat et al., 2019) and climatic conditions using atmospheric reanalysis data (ERA5) in addition to past in-situ measurement data (firn core, GPR and GPS), to explore spectral, thermal, and windward slope relationships with a view towards generating an algorithm for their automatic detection. Moreover, we provide for the first time the first calculation of the absolute megadune movement, and its different components: ice-flow and sedimentological progradation. The analysis of absolute megadune movement has important implications on the remote sensing ice dynamic measurements, in particular on ice flow measurements and elevation changes.".**

*L26 - Evolution and trends of what? albedo, thermal properties, migration, SMB,...*
**A: The sentence has been removed owing to the restructuring of the abstract. We now clarify the aims in the introduction section, see previous comment for details.**

*L50-57 - Consider moving the goal and focus to the end of the introduction*
**A: We have moved the aims of the manuscript at the end of the introduction section as suggested, see comment at lines 25-26 for details.**

*L46 - There is 1:1 overlap with this conference proceeding in this paper. Consider removing this overlap to make the story more focused on the novelty.*
**A: The conference proceedings only presented a very preliminary assessment of megadune mapping, which is greatly expanded on in this study. We have improved the text throughout to focus on the novelty of the manuscript**

*L56 - What is meant by this? What is under process? Who is doing that where and why? Why is that relevant to the paper?*
**A: The data acquired during the EAIIST traverse are still being processed, which is why we cannot report GPS-GPR surveys in the area. We report this information and the website to allow the reader to download the data once they become available.**

*Figure1 - I would find it interesting to see the transects on the images of panels b+c. The figure contains a lot of unused white space. Consider reorganising to show the satellite data in more detail.*
**A: Thank you for the comment. The figure has been reorganized to show the study areas in detail and the transects more clearly.**

*L59 - Impossible to see on the figure*
**A: The figure has been remade and now transects and study areas are visible.**

*L63-68 - Consider moving the aim and summary to the end of the introduction.*
**A: The aim of the study has been moved to the end of the introduction as suggested, see comment at lines 25-26 for details on the new text**

*1.1 Megadune fields - This subsection should be integrated in the introduction with part on megadunes that was mentioned before aim, goal etc.*
**A: The subsection has been integrated in the introduction section as suggested**

*2 Material and method - Rephrase to data and methods. Please add a Study area subsection that describes the area, transects, etc.*
**A: The name of the section has been rephrased to "Data and methods". We have added a subsection 1.1. with the description of the study area as suggested.**

*2.1 Materials - Rephrase to data section. Reorganise the data section with a more clear and complete overview (e.g. subsections for Landsat, Sentinel-2, wind speed, elevation data) now it is all being diluted in one section and being repeated or reintroduced later. I would add the pre-processing steps (e.g. FCC) to the satellite data section*
**A: The section has been rephrased to "Data" and has been reorganized to the following subsections: 2.1 Data; 2.1.1. Satellite datasets; 2.1.2 Atmospheric reanalysis dataset; 2.1.3 Topographic dataset; 2.2 Methods; 2.2.1 Automatic detection of glazed snow surfaces; 2.2.2 Megadune movement estimation**

*L122 - First describe all the data that has been used before explaining all the processing that has been applied on it.*
**A: We have reorganized the data section by clearly separating data and methods**

*L129 - Why only Sentinel-2 for migration and why only band 2 It would make much more logical to use all data for all experiments. Now it reads a bit as a patchwork and it is difficult for the reader to keep overview as many experiments use subtle different data sets and/or pre-processing*
**A: Both Landsat-8 and Sentinel-2 were used to estimate megadune migration. For Sentinel-2, Band 8 NIR was used because megadune are more easily recognized in this band. We now specify this more clearly in the data and methods section, where we have also written why different bands and sensors were used for the different experiments. At lines 134-146, we have**

written: "To map glazed surfaces on megadunes, we used Landsat 8 OLI data as the method relies on the calculation of the albedo, which has been thoroughly validated for Landsat 8 OLI (Traversa et al., 2021a). Additionally, Landsat 8 OLI is available for a longer period of time compared to Sentinel-2, allowing us to investigate temporal evolution of the megadune area. In the megadune area, the difference between snow glazed surfaces and snow is higher considering NIR spectral albedo and brightness temperature (Traversa et al., 2021b). To a "higher" amount of solar radiation absorbed by the glazed surface, corresponds also a different brightness temperature on snow glazed surfaces (Fujii et al., 1987; Scambos et al., 2012 and references therein). In fact, these zones show a higher brightness temperature compared to the upwind part of the dune characterised by the snow surface. In detail, we used Landsat 8 OLI NearIfraRed band (NIR band 5, with a ground resolution of 30 m) to calculate NIR albedo and thermal infrared (TIRS 1) band 10 to calculate brightness temperature (with a ground resolution of 100 m, provided resampled to 30 m). To perform the megadune migration analysis (sect 2.2.2), we used the panchromatic band of Landsat 8 OLI, as this band has a higher resolution (15 m) compared to the other spectral bands of the Landsat. For comparison, Sentinel-2 images were also used, specifically Band 8 NIR (10 m spatial resolution), which allows better observing differences between snow and glazed surfaces compared to the other visible and infrared bands."

*L131 - This should be in a separate subsection.*
**A: We now mention ERA5 data in a separate subsection, 2.1.2, as suggested**

*L141 - Does that mean:*

*$3<x$*
*$5<x$*
*$7<x$*
*$11<x$*
*or*
*$3<x<5$*
*$5<x<7$*
*$7<x<11$*
*$11<x$*
*?*
**A:  We meant $3<x, 5<x$, etc. We have rephrased the sentence to make this clearer.**

*L148 - I guess the mosaic was used and not the individual tiles? If so, please clarify.*
**A: Yes, we mosaicked the REMA tiles. We now clarify this in the manuscript**

*L156 - This is a sentence that does not say anything. Would be better to repeat the aim than to ask the reader to look up the aim (if he/she is reading diagonally)*
**A: The sentence has been removed in the restructuring of the data and methods section**

*L157 - I think it is important to clarify throughout the paper that you discuss thermal brightness temperature (to avoid confusion with microwave BT). This can be done easily by always adding thermal.*
**A: Thank you for the comment. We now always add "thermal" to clarify.**

*L160-162 - Should be part of the data as it provides motivation for data selection.*
**A: we have now moved this sentence to the data section, at lines 126-127 of the revised manuscript, where we have written: " Satellite images from Landsat 8 OLI (supplementary Table A1) were chosen at dates close to the first stripe acquisitions of the REMA DEM (2013, Table A2)."**

*L162-163 - Where is that done in the paper? How?*
**A: The choice of sample transects has now been moved at the start of the data section, at lines 116-120 of the revised manuscript, where we have written: "We further created 5 sample transects in the EAIIST area (Fig. 1c) and visually identified thresholds of albedo, thermal brightness temperature and SPWD to discriminate between glazed surfaces and surrounding snow. The 5 transects were created in different areas of the megadune field and they show relatively different wind directions and topographic aspect and slope, with the aim of representing the widest possible range of SPWD values.". The choice is based on the transects having different values of wind direction, topographic aspect and slope, with the aim of representing the largest possible range of SPWD values. The transects are now shown in detail in Figure 1c.**

*L163-165- Should be part of the study area subsection. Why are 7 transects mentioned when the rest of the paper only mentions one transect*
**A: We now describe the transects directly in the study area subsection. Figure 2 only shows one transects for simplicity, although values mentioned in the text referred to all transects. We now also show scatterplots of SPWD, NIR albedo and brightness temperature along all transects in Figure 3 of the revised manuscript.**

*L165-166 - Should be added to the plots and not here. What is meant by weigthed? How?*
**A: Thank you for the comment. The moving average was weighted based on the distance to the central pixel. This is now written directly in section 2.2.1, at lines 205-207 of the revised manuscript: "Moreover, we determined the strength of the relationship between SPWD vs NIR albedo, and SPWD vs thermal brightness temperature (applied on the moving averages of 11 pixels weighted based on the distance from the central point) using linear regression. "**

*L167-168 - How is done for the different transects? How are they combined?*
**A: Thank you for the comment. Please note we now report this analysis using linear regression. All transect values were used in the analysis, so we compared each SPWD value at location x against the corresponding NIR albedo and brightness temperature at the same location.**

*L 175 - why capital A?*
**A: The title of the section has been changed owing to the restructuring of the manuscript**

*L181-187 - Preprocessing of Landsat data should be moved to data section*
**A: Thank you for the comment. Calculation of the albedo is a full processing chain which is necessary for the mapping of megadunes. We have shortened this section and merged it in section 2.2.1 where we explain how megadune detection was carried out.**

*L188-193 - Preprocessing of Landsat data should be moved to Landsat subsection of data section*

**A: Thank you for the comment. Calculation of the albedo is a full processing chain which is necessary for the mapping of megadunes. We have shortened this section and merged it in section 2.2.1 where we explain how megadune detection was carried out.**

*L191 - As regards is no correct English to start a sentence.*
**A: As the section was restructured, sentences starting with "as regards" have been removed.**

*L210 - Why? Why these bands and not other?*
**A: We apologise for the lack of clarity in the sentence. The FCC was carried out to confirm that NIR albedo explains the highest variance, and allows discriminating the different surfaces, as observed in previous studies (e.g. Frezzotti et al., 2002, Scambos et al., 2012). We have rewritten the sentence to clarify that we used NIR band to perform the sastrugi identification and refer to the aforementioned studies early in the data section.**

*L212-216 - This reads like a lot of hocus pocus without a clear motivation and/or justification. Why are these steps needed? What is their added value? Why were these choices made (and no others)? Was this method benchmarked etc?*
**A: Please see the above comment**

*L216 - What is the methodological principle behind the edge detection? Canny? Sobel?*
**A: It is a Canny edge-detection method. We now state this explicitly in the manuscript**

*L218 - What are small differences? Differencs in what? Reflectance? Migration? Quantify and clarify.*
**A: The sentence has now been removed due to the shortening of the manuscript**

*L220 - how was the interpolation done?*
**A: this was done using bilinear interpolation and is now clarified in the manuscript**

*L223-228 - See major comment on SPWD calculation.*
**A: We have recalculated the SPWD. See major comment for details**

*Figure 3 - Why is there such a large difference in GPS vs REMA on the windward side only? Does that affect any of the subsequent conclusions?*
**A: The difference is caused by the fact that megadunes migrate on the windward side. We have added comments to better explain the megadune migration process in the discussion section 4.2, at lines 430-433 and the conclusion section at lines 480-483. See major comment for details**

*L250-251 - This difference (e.g. with respect to the albedo/thermal analysis) be clarified much earlier in the data section.*
**A: We now clarify at the start of the data section (section 2.1.1 in the revised manuscript) what satellites and bands were used for the megadune mapping and migration analysis. See comment at lines 129 for details**

*L257-259 - Again a different data set over a different period. This is confusing for the reader and should be handled more uniformally in the data section.*
**A: Thank you for the comment. Landsat 7 ETM+ data showed unreliable results and their analysis has now been removed from the manuscript.**

*L259-262 - This is a motivation for the data selection and should be part of the data subsection.*
**A: The sentence has been removed as we have removed this analysis, see comment above.**

*L268 - which previous results? We have not seen results yet.*
**A: We have removed "with previous results" in this sentence**

*L269-270 - See also earlier comment: what is the motivation / benchmark for these methods?*
**A: Please see comment at lines 210 for the detailed answer**

*L273 - Again a different method on a different data set. The readers gets lost and it becomes difficult to compare.*
**A: The analysis of Landsat 7 data has now been removed, see comment at lines 257-259**

*L273-274 - What does that mean?*
**A: We now report the values that were used to perform the cleaning process of the direction raster at lines 240-241 of the revised manuscript: "(angles < 200° and > 240°, intensity < 5 ma$^{-1}$),"**

*L274 - How?*
**A: Thinning refers to reducing the number of cells used to represent the width of the features, in this case to 1 pixel, to aid in vectorization. We now clarify this in the manuscript at lines 241-242.**

*L276-277 - What is the accuracy of this product? Can you see meter displacements in 450m products?*
**A: The average error of Measures Ice flow velocity has been estimated as 3-4% (Rignot et al., 2017). We now report this information in the manuscript.**

*L281-285 - This reads like methodology and should be moved to a methodology section.*
**A: Thank you for the comment. Sentences such as this have been moved to the methods section**

*Figure 4 - Figures are difficult to read (not sharp and small fonts for labels) and lack clear labels to allow the reader to interpret it without reading the caption. Add things like NIR and albedo in large to the titles/labels so the reader should not look up what the subtle differences between the figures is. I don't think that it is necessary to show both original and detrened panels. They seem redundant for the conclusions.*
**A: We now show a lower number of panels, with one with raw topography (to show the megadune morphology) and two detrended panels with NIR albedo and brightness temperature anomaly. We have increased the font size as suggested to ease readability of the figures.**

*L292 - quantify? What is smaller?*
**A: The sentence has been removed to shorten the manuscript**

*L292 - which tendency? It was not explained yet what tendency was.*
**A: The general pattern is of higher values in the upwind area and lower ones downwind for NIR albedo. We now clarify this early on in the results section, at lines 255-257 of the revised manuscript: "On average, in the 5 analysed transects NIR albedo ranges from 0.81 to 0.86 (α) in**

**the upwind area (snow sastrugi) and from 0.73 to 0.81 (α) downwind (glazed surfaces), with differences inside the transects of about 0.07 (α) with a maximum value of 0.1 (α). "**

*L293-295 - Sorry, but I do not really understand this sentence. Consider rephrasing. What heterogeneity? What exponential effect?*
**A: We have removed this sentence, as we now consider only scenes with a SZA < 75°, see your major comment.**

*L293 - which small difference? Be specific.*
**A: We have removed the paragraph on the differences between scenes with higher/lower SZA, see previous and major comment**

*L295 - Which first/last three? Be specific*
**A: Please see comment above**

*L298 - ...is 0.03-0.04 lower than in the upwind zone...*
**A: The sentence has been removed owing to the restructuring of the manuscript.**

*L307-309 - This is not shown in Figure 2. Figure 2 show false colour images and there you can easily discriminate them.*
**A: The sentence has been removed in the shortening of the manuscript**

*L310-312 - Should be part of the method and not being introduced in the results.*
**A: We have removed scenes with SZA > 75°, see previous comments about this**

*L313-315 - Appendix figures should not be essential to interpret the main paper. If the reader needs to look at them, they should be part of the main paper.*
*A:* **We have removed scenes with SZA > 75°, see previous comments about this**

*L313-320 - Based on this difference I would consider only using images with SZA<70 throughout the paper. This will make the interpretation/flow of the paper much more logical.*
*A:* **We have removed scenes with SZA > 75°, see previous comments about this**

*L322 - Should be moved to methods section.*
**A: We have removed this sentence as this information was already in the methods section**

*L326 - what is a farthest date?*
**A: We meant the date farthest from the summer solstice. We have rephrased the sentence as: "The difference increases on the date farthest from the solstice, 28-Jan-2014" to clarify this.**

*Figure 5- Same comment as Fig.4:*
*-make figure sharper (pdfs are better for line plots)*
*- increase text size*
*- add labels/titles (e.g. Band 10, Band 11)*
**A: Thank you for the comment. We have remade all figures to make them more readable**

*L363-364 - This seems an artefact of the method and should require a better way to calculate SPWD that also accounts for subtle differences in wind direction.*

**A: We have recalculated the SPWD as suggested, see major comment for details**

*L368-369 - I guess these values highly depend on the methodology and should be recalculated with wind/slope directions that have better resolution than 22.5 degrees.*
**A: We have recalculated the SPWD as suggested, see major comment for details**

*L371 - Should be part of the methodology*
**A: The sentence has been removed in the shortening of the manuscript**

*L371-379 - I don't think this analysis adds a lot. It is already clear from the previous figures that the variables correlate so what is the added value of knowing which one correlated "better".*
**A: The added value is to identify whether NIR albedo, brightness temperatures and SPWD can be used together to perform a classification of megadune areas. We now report our aims more clearly in the manuscript.**

*L385-388 - This should be part of the methodology, not result. How is this classification done? Which methodology? Much information (that should be part of the methodology) is missing here.*
**A: Details about the methodology employed to detect snow glazed surfaces are now included in the methods section, see major comment to Figure 7**

*Figure 7 - What do we see in this figure? What are the colors (legend?)? What are the purple dashed areas? What are the two dates on the bottom? At the moment this figure lacks all context to read it.*
**A: The figure has been remade taking into account suggestions by both reviewers. Please also note we now show the detection of snow glazed surfaces with/without SPWD on the entire Landsat tile and when using threshold identified on the entire Landsat tile and on a narrower area, see major comment to Figure 7.**

*L407-409 - Should be part of the methodology*
**A: The sentence has been removed as the information is already in the methods section**

*L414 - Does that mean we cannot draw conclusion from it? Please clarify*
**A: Yes, feature tracking using the two different satellite sensors together was unreliable and has been removed from the study**

*L425 – 427 - This creates a potential problem: see also earlier comment.*
**A: We have added comments in the discussion and conclusion sections, at lines 430-433 and 480-483 of the revised manuscript, to better explain the megadune migration process. See major comment for the full text.**

*L441 - ERA5 might also be far from perfect! It is a reanalysis dataset: not reality*
**A: We have added: "as well as inaccuracies in the ERA5 wind direction" here to clarify the possible errors in ERA5.**

*L444 - This cannot be concluded on 8 cell SPWD (see earlier methodological comment)*
**A: We have now recalculated the SPWD. See major comment for details**

*L448-450 - This should be addressed in this study with a better SPWD method and not be postponed for later work.*
**A: Please see comment above**

*L452 - Where did you show this? I have never seen that in the paper?*
**A: We have removed this sentence**

*L454-456 - The methodology for this approach is completelt lacking. Moreover it*
**A: Methodological details on the detection of snow glazed surfaces are now provided in the methods section, please see major comment to figure 7 for details.**

*L472-493 - These are all results and should be part of the results before the discussion section.*
**A: The information in this section has been moved to the results section 3.1 as suggested**

*L547 - I would advice to deposit the data corresponding to the paper in a open access repository (with doi) and not rely on requests to the corresponding author.*
**A: The data will be placed in an open repository as suggested**

---

## Author Response (AR2)

**AUTHOR COMMENTS**

The authors thank the Reviewer and Editor for their constructive comments and corrections that have increased the scientific quality of the manuscript and its clarity.

Here we present our answers to the reviewer's comments. In particular, the manuscript has been mainly modified in the classification method for glazed snow detection. Now, the methodology is more objective, based on a supervised classification and it is provided with an accuracy assessment. In addition, we now display a comparison between different combinations of parameters in order to define the best approach to automatically detect glazed surfaces. Finally, we expanded and better underlined the impacts and key points of the manuscript in the introduction, discussion and conclusion sections.

We hope that the revised version of the manuscript has improved the quality of the text and of the scientific message.

Changes and answers in response to the Reviewer's comments/suggestions (in italic) are highlighted in bold.

Major comments:

*- Impact: the authors claim in the abstract that their "results present significant implications for Surface Mass Balance estimation, paleoclimate reconstruction using ice cores and for the measurements using optical and radar images/data in the megadune area." but only in the conclusion clarify what these implications are. I think it is important to highlight this much more in the discussion which is currently mostly a technical discussion where it if difficult to extract the main messages.*

**A: we have now modified the introduction and discussion sections, especially in the subsection regarding the automatic detection of glazed snow, better highlighting the impact of the manuscript on the detection of glazed surfaces using "morphological" parameters (SPWD) and the relevance of megadune migration. Atmospheric circulation models used to calculate SMB are not able to reproduce the ice sheet ablation areas (megadunes and glazed surface areas) that represent a negative value close to the total error reported for the Mass Balance of East Antarctica. In the present study, we have also demonstrated the importance of SPWD on the detection of glazed surface/ablation areas. For the first time, our manuscript measured the migration of the megadunes in all their components, previously hypothesized using only GPR surveys, and pointed out the significant difference (in direction and value) between the detection of surface velocity using optical and inSAR techniques. These new results represent a new ground truth and foundation of knowledge for ice sheet mass balance research, in particular for satellite altimeter and ice velocity derived by remote sensing measurements (e.g., radar vs optical/lidar). Moreover, the measurements, in all components (absolute, upgradation, ice flow), of megadune migration and their burying process (300 yrs) provide information about the distortion of reconstruction of paleoclimate based on firn/ice cores drilled in megadune or downstream areas.**

*- Rigour of the classification method: Traversa et al present a method to classify glazed snow which is based on ad-hoc arbitrary thresholds (which could explain the large differences depending on the used method etc). The development/implementation of this method should be done much more rigorously with some sort of optimization (e.g. implementing a trained classifier instead of subjective thresholds) and a sensitivity analysis showing the sensitivity of the method to the used thresholds etc. Currently, it is a subjective analysis which cannot be trusted.*

**A: we now propose a comparison of the previous self-defined-threshold method with a more objective approach, i.e., supervised classification. In both methodologies, we evaluated different combinations of key parameters (i.e., NIR albedo, thermal BT and SPWD) and in all the cases, we assessed the accuracy of the method in order to define which approach is the most suited to detect glazed snow surfaces. This analysis was performed both on the scene from 17-Dec-2015 (the highest quality image of the dataset) and on the entire 2013-2014 season (four scenes). The methods, results and discussion sections have been changed accordingly; see lines 212-226, 317-338, and 402-432 of the revised manuscript.**

Specific comments:

*- L19-20: the classification with and without SPWD should be done much more rigorously with an automated classifier that allows to assess feature importance.*

**A: we now propose a comparison of the previous method with a more objective approach, i.e., supervised classification (see answer to major comment above). The role of SPWD is now more evident and the accuracy can be better quantified. Therefore, the sentence has been modified to: "These parameters allowed us to characterise and perform an automated detection of the glazed surfaces, and we determined the influence of the SPWD by evaluating different combinations of these parameters. The inclusion of the SPWD significantly increased the accuracy of the method, doubling it in certain analysed scenes." (lines 18-21 of the revised manuscript).**

*- L23 "significant implications" these implications are barely handled in the paper (not in results nor discussion) and should be stressed much clearer.*

**A: we stressed the implications of our study on SMB, by adding the following sentence: "In conclusion, the detection of glazed surface/ablation areas by satellite images is challenging because of differences in illumination and meteorological conditions (cloud cover, blowing snow etc.) among different satellite images. Nevertheless, the high resolution digital terrain model (REMA) allows to calculate a SPWD with unprecedented detail, similar to the resolution of available optical satellites (Landast 8-9, Sentinel), and significantly improves the detection of glazed/ablation surfaces at ten-meter scale across the continent; therefore, it could significantly improve research on the SMB of the Antarctic Ice Sheet." (lines 428-432 of the revised manuscript).**

*- Fig 2+3+5 look like poor quality screenshots of excel. Figure quality should be improved before it can be published*

**A: figures 2-3-4 (previously figure 5) have been now improved in quality and exported at 300 dpi.**

*- Fig1a+b: this figure uses a diverging color palette whereas data are continuous (not diverging). Check https://www.nature.com/articles/s41467-020-19160-7 for advice and replace it with correct colors.*

**A: the figure has been modified according to the suggested paper using the correct colours.**

*- L208: pluriannual -> I think multi-annual is a more common term*

**A: the sentence has been modified as suggested**

*- L214-219: this methodology should be expanded to a more objective classification technique (e.g. based on supervised classification + optimization) instead of ad-hoc subjective thresholds*

**A: we now include a supervised classification with accuracy assessment (see answer to the second major comment for details)**

*- L222: "difference between topographic slope and SPWD. " seems like a ghost sentence*

**A: we have removed this sentence as suggested.**

*- L221: I miss a clear identification of the differences between the submitted paper and the Frezzotti paper. The main topic of this paper is megadune migration so it show much better the differences.*

**A: The analysis by Frezzotti et al. was based on the comparison between sedimentary structure from GPR and surface morphology. Here, the movement is calculated and then quantified by feature tracking of surface morphologies from satellite image pairs (Landsat and Sentinel-2) and comparison of surface topography (1999 traverse) with the REMA DEM (2014). The new modified sentence is: "Frezzotti et al. (2002b) and Ekaykin et al. (2015), based on the sedimentary structure of buried megadunes (using GPR and GPS), pointed out that the megadune migration and ice sheet surface flow show a similar intensity but opposite directions and that megadunes migrates upwind with time, burying the glazed surface of the leeward face. Here, by using different satellite images and field data, we are able to provide and quantify megadune migration components…" (new lines 228-231).**

*- L255 "5": as a rule of thumb numbers below 10 should be written as text in the main text*

**A: we have replaced the number with text as suggested**

*- Fig. 2a+b would be clearer if homogeneous. A) shows non-normalized data while b) shows normalized data.*

**A: we have modified the figure as suggested**

*- L289 "in accordance with previous authors", if it is completely in accordance, then I wonder what the novelty/ added value is.*

**A: this part of sentence was removed since none of the cited papers had analysed NIR and SPWD in relation with megadune flanks.**

*- L290-294: I find it very difficult to see significant differences in variability between glazed on non-glazed surfaces, so I think this needs to be proven quantitively to avoid it is a subjective interpretation.*

**A: we have quantified the differences, finding that the variability of NIR albedo more than doubles between the two flanks of the dunes (0.3% vs 0.7%). The sentence was rephrased as follows: "Based on the transects, the variability in NIR albedo at seasonal (spring-summer) to pluriannual scale is observed to be twice as large in the snow accumulation area on the upwind flank and the bottom of the leeward flank (Fig. 2), compared to the glazed surface NIR albedo (0.7% compared to 0.3% NIR albedo variability), which remains more stable and more highly correlated at seasonal (spring-summer) and pluriannual scale" (lines 295-298 of the revised manuscript).**

*- L312-317: I don't think this can be concluded based on a subjective classification algorithm*

**A: we now include a supervised classification with accuracy assessment (see answer to second major comment for details), which has made the methodology more objective.**

*- Fig. 5b: the GPR profiles are impossible to interpret (especially with the dashed lines) and it is not clear what they add to the story*

**A: we have removed the GPR profiles from the image**

*- L364-369: again not possible to interpret based on subjective analysis*

**A: we now included a supervised classification with accuracy assessment (see answer to second major comment for details), which has made the methodology more objective.**

*- L371-394: I found it difficult to find an impactful result in this part of the discussion*

**A: we have modified and shortened this part of the discussion, in order to highlight only the relevant aspects.**

*- Section: I think this section gets lost in the details and fails to convey a clear take home message*

**A: it is not clear what section the comment refers to, since the reference is missing. Nevertheless, both the discussion and the conclusion were strongly modified in order to highlight the relevant aspects.**

---

## Author Response (AR3)

AUTHOR COMMENTS

The authors thank the Reviewer and Editor for their constructive comments and corrections that have increased the scientific quality of the manuscript and its clarity.

Here we present our answers to the reviewer's comments. In particular, in addition to minor technical corrections, we included the comparison with the dissertation of Courville (2007), which we found significant in the context of the manuscript. Furthermore, the Figure 1-2-3 were modified in accordance with colour-blind standards.

We hope that the revised version of the manuscript has improved the quality of the text and of the scientific message.

Changes and answers in response to the Reviewer's comments/suggestions (in italic) are highlighted in bold.

*Line 64, should be "Dadic et al. (2013) based their analysis" (and also, Ruzica Dadic, the author of the paper is a she, so if you want to use the singular, it should be "based her analysis")*

**A: the sentence was modified as suggested.**

*Line 70: what is this comparing to, i.e., what is on average higher? Temperatures at depth vs. the snow surface? Spectral differences also lead to an effect on temperatures, which is on average higher than on the snow surface (Fujii et al., 1987).*
**A: the comparison was intended to be between glazed surfaces and surrounding snow; thus, the sentence was modified as follows: "Spectral differences also lead to an effect on temperature, which is on average higher over glazed surfaces than on the snow surface (Fujii et al., 1987)."**

*Line 92: in this particular megadune region? Or across all megadune regions?*
**A: the megadune region from Palm et al. (2019) included the study areas of the present manuscript. The sentence was then modified as follows: "…megadune region (Palm et al., 2019), that includes the study areas of the present work.**

*Line 117 and figure 1 (boxes c and d): It's confusing as stated that the wind direction in both boxes c and d are coming from the SW due to the orientation differences in both boxes. I think a directional/south arrow would clear that up in boxes c and d (even though it is clear from box b what the orientation is, just think a directional arrow would help)*
**A: north arrows were added to boxes c and d of the figure 1, which now is as follows:**

[Figure]

*Line 121: DEM should be defined in the first instance it is used (is defined later on)*
**A: the sentence was modified as suggested and now DEM is defined at line 121 and not later on.**

*Line 130: It would be good to mention the size of the imagery/scenes used in the analysis, since that factors into the number of features listed in Table 1.*
**A: the size of both L8 and S2 tiles is now mentioned at lines 130-131 ("Landsat 8 OLI satellite images (tile area ~38,000 km$^2$) and Sentinel-2 images (tile area ~12,000 km$^2$"), as well as in the caption of table 1 ("Table 1. Results of the absolute migration of megadunes calculated from IMCORR based on Landsat 8 OLI (L8), with tile area of ~38,000 km2, and Sentinel-2 (S2), with tile area of ~12,000km2, imagery at the It-ITASE and EAIIST sites.").**

*Line 145: This sentence is confusing as written: To a "higher" amount of solar radiation absorbed by the glazed surface, corresponds also a different BT on snow glazed surfaces (Fujii et al., 1987; Scambos et al., 2012 and references therein).*
*I think it means: A "higher" amount of solar radiation is absorbed by the glazed surface, and also corresponds to a different BT on snow glazed surfaces (Fujii et al., 1987; Scambos et al., 2012 and references therein).*
**A: the sentence was modified as suggested.**

*Line 221: How were these threshold values chosen? I.e., what does it mean that a "conditional calculation" was used to automatically map the snow. This should be explained in more detail since it is a critical step in determining glazed area extent.*
**A: the values were chosen based on the specific case of the analysed images. This concept and the definition of conditional calculation are now better explained in the manuscript at lines 220-222 of the revised version. The new sentences are as follows: "For the self-defined-threshold method, we applied a conditional evaluation (i.e., output result for each pixel based on whether the pixel value is assessed as true or false in a set conditional statement) to automatically map glazed snow. The thresholds were visually identified and iteratively adjusted to obtain a best fit as follows: SPWD > 1 m km$^{-1}$, with the aim of considering the leeward flanks only, NIR albedo < 0.82 and thermal BT > 246.5° K.".**

*Table 1: as mentioned in the remark on for the paragraph discussing satellite imagery, it seems like the number of features is related to the coverage area of the satellite imagery, or is it all due to the higher resolution? If that's the case, I think the different coverage of the imagery is worth mentioning, or if there is another cause of the variation besides the resolution, that should be mentioned here.*
**A: see comment above and modification to the manuscript referring to this issue. A further sentence was added at line 348, i.e., "…even if the number of pixels is higher (with a ratio of 1.4) in Landsat scenes.".**

*Line 290: Figure 2: Looks like some of the figure caption is incomplete or mislabeled, i.e., there is a missing description of box d. "Corresponding normalised moving average of NIR 290 albedo (b) and thermal BT TIRS1 (c) during the austral summer season 2013-2014 for transect C and elevation from REMA DEM (detrended topography)." Isn't the description in the caption*

*describing (c) really describing the normalized data in box (d)? Regardless, this caption should be clarified.*

**A: the caption of figure 2 was modified and made clearer as follows: "Figure 2: Moving averages (based on 11 transect pixels) of NIR albedo (a) and thermal BT TIRS1 (c) between November 2013 and February 2014 for transect C at the EAIIST site (see Fig. 1c for location) and elevation from REMA DEM. Corresponding normalised moving averages of NIR albedo (b) and thermal BT TIRS1 (d) during the austral summer season 2013-2014 for transect C and elevation from REMA DEM (detrended topography)."**

*Line 273: Dic should be Dec*

**A: the sentence was modified as suggested.**

*Line 274: "Similar" should not be capitalized*

**A: the sentence was modified as suggested.**

*Line 513: I have data in my dissertation (Courville 2007) not published in a journal article based on migration rates calculated by Mark Fahnestock based on satellite remote sensing images and AVHRR data (from Fahnestock et al., 2000, but updated and not published) and firn cores that were drilled in 2003/04. I determined a burial rate of 330 years based on our firn core, and migration rates of approx. 12 m yr-1 based on a more detailed analysis of the features mentioned in Fahnestock et al. 2000. This data, not published outside of my dissertation, is obviously not well known and the authors wouldn't be expected to know about it or cite it. I just found it very interesting that the values are so close to one another in such different spots.*

**A: the comparison, as very inherent to the present analysis, was added to the manuscript. Now, at lines 462-465 the following sentence was added: "These results are strongly in accordance with Courville (2007), who determined a burial rate of 330 years based on a firn core drilled in 2003/04, and migration rates of approx. 12 m a-1 (from AVHRR data) at a field located at 80° 47' S, 124° 29' E in the megadune region of EAIIST site."**

---

## Author Response (AR4)

AUTHOR COMMENTS

The authors thank again the Reviewer for his constructive comment that has increased the clarity of the manuscript.

The missing blank in table 1 was added and we hope that the revised version of the manuscript is now adequate for the publication. In the text file, where no figures are shown, the captions of figures and tables are at the end of the manuscript. Let us know if there are further corrections to do.

Thanks again for letting us publish on TC! Merry Christmas!